# Single dose of intravenous miR199a-5p delivery targeting ischemic heart for long-term repair of myocardial infarction

Yu Chen[1], Shuai Liu[2], Yunsong Liang[1], Yutong He [1], Qian Li[1], Jiamian Zhan[1], Honghao Hou [1,3] ✉ & Xiaozhong Qiu [1,3] ✉

Long-term treatment of myocardial infarction is challenging despite medical advances. Tissue engineering shows promise for MI repair, but implantation complexity and uncertain outcomes pose obstacles. microRNAs regulate genes involved in apoptosis, angiogenesis, and myocardial contraction, making them valuable for long-term repair. In this study, we find downregulated miR-199a-5p expression in MI. Intramyocardial injection of miR-199a-5p into the infarcted region of male rats revealed its dual protective effects on the heart. Specifically, miR-199a-5p targets AGTR1, diminishing early oxidative damage post-myocardial infarction, and MARK4, which influences long-term myocardial contractility and enhances cardiac function. To deliver miR-199a-5p efficiently and specifically to ischemic myocardial tissue, we use CSTSMLKAC peptide to construct P-MSN/miR199a-5p nanoparticles. Intravenous administration of these nanoparticles reduces myocardial injury and protects cardiac function. Our findings demonstrate the effectiveness of P-MSN/miR199a-5p nanoparticles in repairing MI through enhanced contraction and anti-apoptosis. miR199a-5p holds significant therapeutic potential for long-term repair of myocardial infarction.

Myocardial infarction (MI), commonly known as a heart attack, is a significant contributor to morbidity and mortality worldwide[1]. The primary goal of MI treatment is to minimize myocardial damage, restore normal cardiac function, and prevent heart failure[2]. Early treatment of MI aims to primarily reduce apoptosis in myocardial cells. Apoptosis, characterized by the activation of caspases, excessive production of ROS, DNA fragmentation, and mitochondrial dysfunction[3], plays a significant role in the progression of MI[4]. Various therapeutic interventions, such as reperfusion strategies and pharmacological agents, have been implemented to promote cell survival and limit apoptosis during the early stages of MI[5]. However, solely inhibiting cell apoptosis may not be sufficient to effectively improve myocardial contractile dysfunction after myocardial infarction, as contractile failure is frequently an expression of adaptive responses by viable myocytes to various forms of non-lethal damage, leading to the survival of cardiomyocytes but impaired function[6,7].

In the later stages of MI, adverse ventricular remodeling characterized by fibrosis, hypertrophy, and changes in ventricular geometry occurs. This remodeling process leads to impaired contractility and increased risk of heart failure. Late-stage treatments primarily focus on enhancing myocardial contractility and preventing adverse remodeling[7,8]. Impaired contractility leads to reduced ejection fraction and compromised hemodynamics[9]. Strategies aimed at enhancing myocardial contractility have shown promise in experimental models, including the use of positive inotropic agents or gene therapy approaches targeting contractile proteins[10]. A therapeutic dilemma is that the use of positive inotropic agents, such as catecholamines or phosphodiesterase-inhibitors, their long-term use is associated with

[1]Guangdong Provincial Key Laboratory of Construction and Detection in Tissue Engineering, Department of Anatomy, School of Basic Medical Sciences, Southern Medical University, Guangdong, Guangzhou, China. [2]The Fifth Affiliated Hospital, Southern Medical University, Guangzhou, Guangdong, People's Republic of China. [3]These authors contributed equally: Honghao Hou, Xiaozhong Qiu. ✉e-mail: ss.hhh89@hotmail.com; qqiuxzh@163.com

increased mortality, possibly due to their potential to exacerbate cell death or promote adverse remodeling[11]. However, finding a balance between optimizing contractility without exacerbating cell death or remodeling remains an ongoing challenge.

Therefore, it is evident that innovative therapeutic strategies are needed to combine early-stage anti-apoptotic treatments with late-stage contractility enhancing strategies. How to uncover the intricate interplay and bidirectional regulation of these two variables within an entire heart is critical for developing effective long-term MI therapies. Recent researches has provided growing evidences for the potential of using miRNA (microRNA), a kind of small non-coding RNA molecules[12,13], as therapeutic agents for MI due to their ability to regulate gene expression and modulate cellular processes involved in cardiac development, apoptosis regulation, angiogenesis, and contractile function[14,15]. MiRNAs can simultaneously regulate multiple target genes[16], allowing for a comprehensive modulation of key cellular processes involved in MI. Selecting specific miRNAs, which can simultaneously inhibit apoptosis and enhance myocardial contractility, offers a promising approach for long-term MI treatment.

Although miRNA-based therapeutics show promise in simultaneously targeting multiple pathways involved in MI, several challenges need to be addressed for their successful implementation[17]. One of the challenges is the efficient delivery of miRNAs to the ischemic myocardium[13]. The stability and biodistribution of miRNAs need to be optimized, and effective delivery systems need to be developed to ensure the miRNAs reach their target cells. Another challenge involves selecting the appropriate combination of anti-apoptotic and contractility-enhancing therapies for maximum therapeutic benefit. The choice of miRNAs and their target genes should be carefully evaluated to achieve desired outcomes without interfering with normal cardiac function or causing adverse effects[18]. Understanding the intricate molecular mechanisms underlying apoptosis and contractility regulation in the context of myocardial infarction is essential to identify the most effective combination of therapeutic agents. Furthermore, it is important to evaluate the long-term effects of miRNA-based therapies and targeted nanocarriers on cardiac remodeling and overall cardiac function. Further preclinical and clinical studies are required to assess the safety, efficacy, and durability of these therapeutic approaches and their potential for translation into clinical practice.

In this study, we observed a decrease in miR199a-5p expression in the infarcted region within 4 weeks of MI. To investigate its potential protective role, we performed intracardiac injection of miR199a-5p mimics and found a reduction in MI-induced cardiac injury and preservation of cardiac function. Subsequently, we aimed to validate the hypothesis that miR199a-5p plays a protective role in the heart during MI injury by regulating ROS flux and myocardial contractility. Our research demonstrates that miR199a-5p specifically targets AGTR1 and MARK4, showing a high degree of conservation across multiple species. Based on these results, we developed an effective miRNA-based delivery platform to reprogram ischemic cardiomyocytes after MI, aiming to enhance contractility and prevent apoptosis. To achieve this, we utilized CSTSMLKAC peptide (PEP), a polypeptide that has demonstrated remarkable selectivity for ischemic myocardium screened by phage demonstrations[19]. In a mouse ischemia model, intravenous injection of exosomes or mitochondria fused to PEP exhibited preferential binding to ischemic cardiac tissue over normal cardiac tissue and control organs[20,21]. Considering the favorable biological protection and delivery efficiency of miRNA, we selected mesoporous silica nanospheres (MSNs) as carriers[22]. We modified the surface of MSNs with PTP peptides and coated them with cationic polyethyleneimine (PEI). The MSNs were then able to interact with miRNAs through electrostatic attraction, resulting in the successful preparation of P-MSN/miRNA nanoparticles. Finally, we evaluated the efficacy of P-MSN/miR199a-5p nanoparticles through tail injection in a rat MI model. Our results demonstrate that this intervention leads to a reduction in cardiomyocyte apoptosis after infarction and improves post-infarction myocardial contractility and cardiac function. In conclusion, our study identifies miR199a-5p as a potential therapeutic tool for long-term repair of infracted myocardium and highlights the effectiveness of P-MSN/miR199a-5p nanoparticles as the nano-drugs for targeted therapy of MI.

## Results

### miR199a-5p is dysregulated in diseased hearts

Analysis of miRNAs expression at different times of myocardial infarction in the GEO database revealed that disrupted expression of miR199a-5p and miR-140-5p widely present in infarcted hearts (Fig. 1a). Either decreased or increased expression levels appeared in infarcted hearts, supporting the notion that these miRNAs play a role in the pathophysiological mechanism of myocardial infarction.

To verify the expression changes of miR199a-5p and miR-140-5p in MI, we examined their expression in rat models of MI. A decrease in miR199a-5p expression was detected in both blood (Fig. 1b) and infarcted myocardium (Fig. 1c) within four weeks after the onset of MI. However, the miR-140-5p expression did not change in blood at 2 and 3 weeks (Fig. 1d) and in infarcted myocardium at 2 and 4 weeks (Fig. 1e) after MI. Additionally, the distribution of miR199a-5p in both cardiomyocytes and cardiac fibroblasts was investigated, revealing enrichment specifically in rat cardiomyocytes. (Fig. 1f). These results suggest that the role and expression of miR199a-5p are highly correlated with MI, but those of miR140-5p are not the focus. To investigate the dynamic expression pattern of miR199a-5p during myocardial infarction progression, we conducted an analysis of miR199a-5p changes at various time points post-infarction within the GEO database. Our examination revealed a consistent decrease in miR199a-5p levels at 6 hours (GSE24591), 1 day (GSE138141), 5 days (GSE53211), and 1 week (GSE114695) following infarction. Notably, divergent trends were observed at 4 weeks and 8 weeks, with miR199a-5p exhibiting a decrease at 4 weeks (GSE124545) and 2 months (GSE18129), while showing an increase at 4 weeks (GSE208159) and 8 weeks (GSE114695). We postulated the presence of individual variations in miR199a-5p alterations between 4 and 8 weeks post-infarction.

To explore the expression changes of miR199a-5p in cardiac diseases other than myocardial infarction, we analyzed its expression in the blood of patients with dilated cardiomyopathy (DCM), tetralogy of Fallot (TOF), and coronary artery disease (CAD), as well as in the hearts of rats with ischemia/reperfusion (I/R) injury using the GEO public database. Our analysis revealed a decrease in miR199a-5p expression in the blood of patients with DCM and TOF, as well as in the hearts of perfused I/R rats. However, in the blood of patients with CAD, although the mean value of miR199a-5p expression was reduced, it did not reach statistical significance (Fig. 1g). These results collectively demonstrate the dynamic modulation of miR199a-5p expression in various heart diseases.

### Long-term MI repair effect after intramyocardial injection of MSN/miR199a-5p

To investigate the function of miR199a-5p in the heart, MSN package miR199a-5p mimics were directly injected into the hearts of rat models of myocardial infarction. The average diameter of the MSN/miR199a-5p particles was measured to be 165 nm. Myocardial infarction and heart failure were induced by permanent ligation of the left anterior descending (LAD) coronary artery in adult rats. Cardiac function was monitored long-term on miR199a mimic-treated rats (Fig. 2a). The results showed that miR199a-5p-mediated cardioprotection was primarily observed during the early phase (2 weeks) after myocardial infarction, with moderate protection in the late stage (6 months or 12 months). Approximately 40% of rats injected with the control miRNA mimic died from heart failure within two weeks after

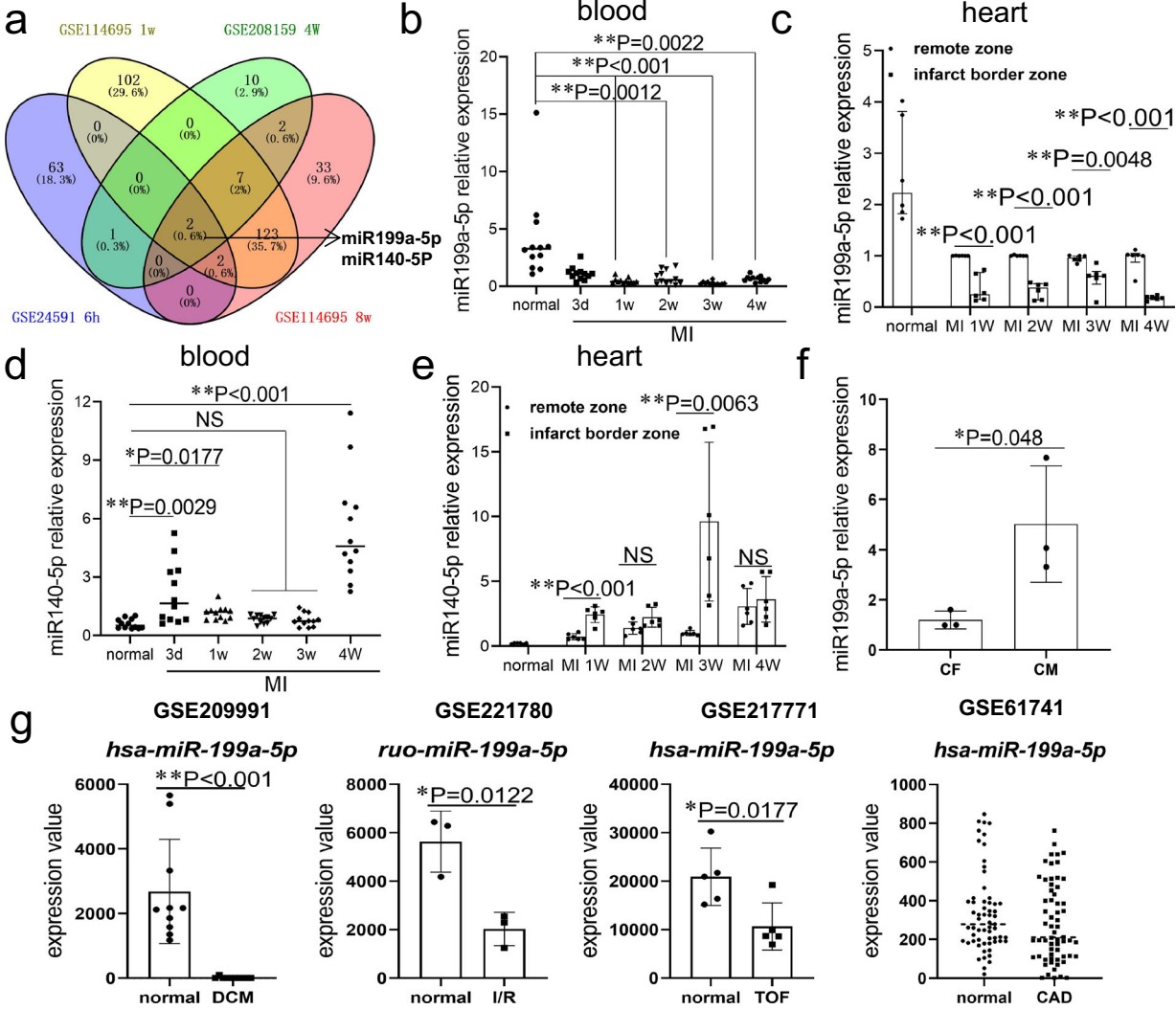

**Fig. 1 | Expression of miR199a-5p in heart diseases and cardiomyocytes.**
**a** Intramyocardial miRNA expression was analyzed during different periods of myocardial infarction to identify miRNAs that affect the pathophysiology of infarction. The numbers in the Venn diagram indicate the number of miRNAs belonging to the corresponding intersection. **b–e** In the rat myocardial infarction (MI) model, qRT-PCR was performed to detect the expression of miR-199a-5p in the blood (**b**, n = 12 rats) and heart tissue (**c**, n = 6 rats) at specified time points post-MI. Additionally, qRT-PCR analysis was conducted to assess miR-140-5p levels in blood (**d**, n = 12 rats) and heart tissue (**e**, n = 6 rats) at designated time points following myocardial infarction. **f** In isolated adult cardiomyocytes and cardiac fibroblasts, qRT-PCR detection of the expression of miR199a-5p, n = 3 rats. **g** Expression

analysis of miR199a5p in the heart or blood of dilated cardiomyopathy (DCM, n = 10 participants), ischemia in perfusion (I/R, n = 3 rats), tetralogy of Fallot (TOF, n = 5 participants), and coronary artery disease (CAD, n = 61 participants) using public Gene Expression Omnibus (GEO) (https://www.ncbi.nlm.nih.gov/geo/) data. The data for (**b**)–(**e**) are presented with median and interquartile range values. Group comparisons for (**c**) and (**e**) were conducted using the Mann-Whitney U test, while multiple independent samples for (**b**) and (**d**) were assessed using the rank sum test. Data for (**f**) and (**g**) are expressed as mean ± standard deviation. The p-values were generated using an unpaired two-tailed Student's t-test. *P < 0.05, **P < 0.01, NS, not significant. Source data are provided as a Source Data file.

myocardial infarction, whereas only about 20% of rats injected with the miR199a-5p mimic succumbed to it (Fig. 2b). Consistent with a previous study reporting a high mortality rate in MI[23], cardiac function assessed by echocardiography showed progressively deteriorating function in control rats but significantly preserved fractional shortening (FS%) in miR199a-5p-treated rats throughout the follow-up period (from week 4 to month 12) (Fig. 2c, detailed data are shown in Supplementary Date 1). These findings demonstrate the cardioprotective role of miR199a-5p in myocardial infarction.

Myocytes that survive a myocardial infarction undergo eccentric hypertrophy, which contributes to adverse cardiac remodeling and systolic dysfunction. The main clinical manifestations of this process include cardiac hypertrophy and pulmonary edema[24]. Furthermore, we conducted a systematic analysis of the cardiac phenotype in rats injected with miR199a-5p mimics at 6 and 12 months after infarction.

The miR199a-5p-treated rats exhibited lower heart weights and lung weights compared to the control group, as evidenced by reduced heart weight/tibial length, heart weight/body weight and lung weight/body weight ratios (Figs. 2d, e and S1a, b). The expression of miR199a-5p in the infarct zone was increased and sustained for 1 year after injection of MSN package miR199a-5p mimics compared with control (Fig. S1c). Echocardiographic measurements revealed that the rats injected with miR199a-5p after myocardial infarction had shorter left ventricular internal dimensions in diastole (LVIDs) compared to the controls (Fig. 2f, g). Additionally, there was a decrease in serum protein expression of brain natriuretic peptide (BNP), a marker of cardiac function, at both 6 and 12 months post-miR199a-5p injection (Fig. 2h, i). Histological analysis confirmed a reduction in cardiac infarct size following injection of miR199a-5p mimics (Fig. 2j, k) and no side effects on other organs (Fig. S1d). Importantly, these results

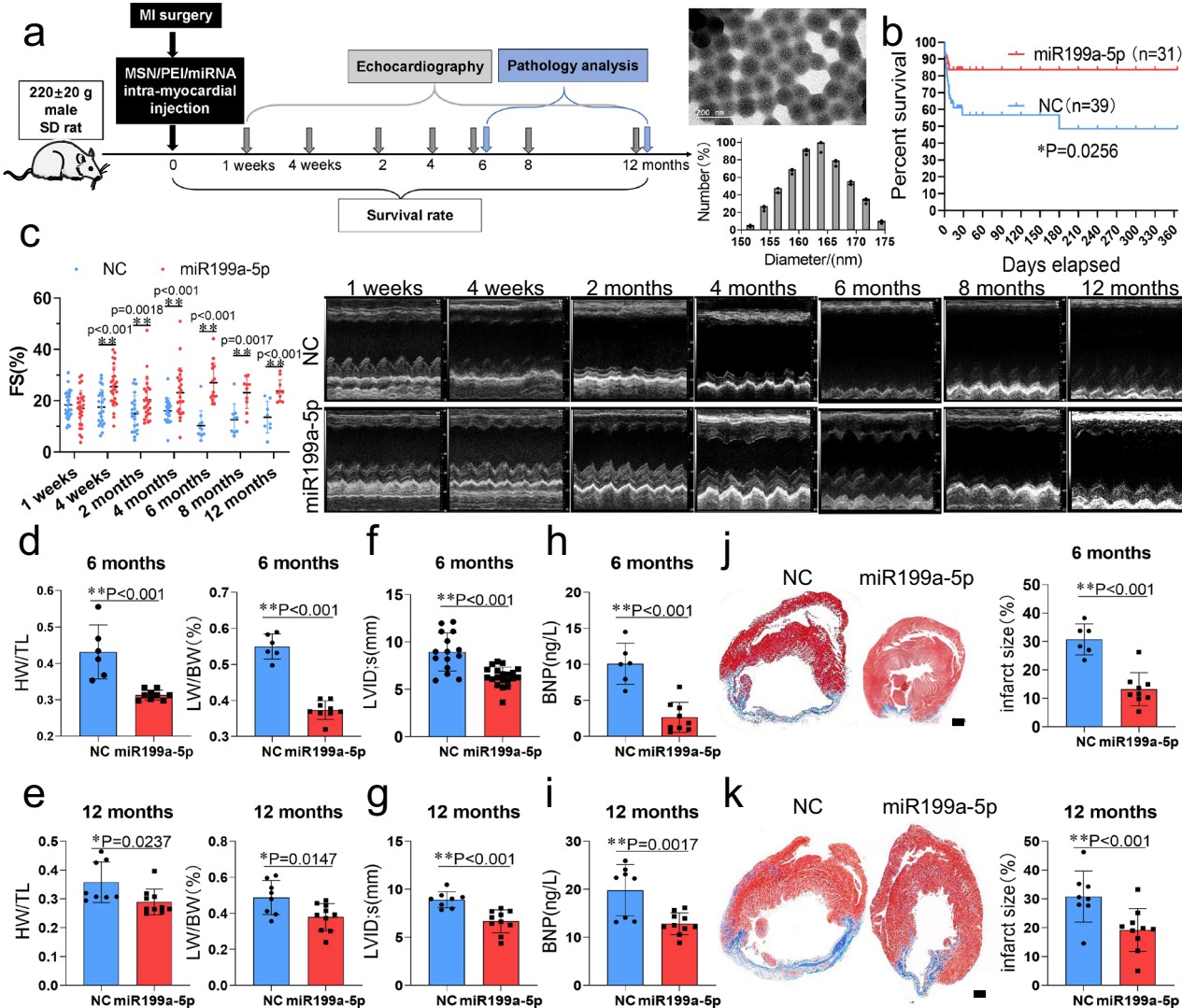

**Fig. 2 | Direct injection of miR199a-5p mimics protects the heart from myocardial infarction. a** Experimental design: Assessment of short- and long-term post-MI cardiac function and morphology in rats receiving MSN package miR199a-5p mimics. Transmission electron microscopy photographs (scale bars: 200 nm) and percentage size distribution of MSN/miR199a-5p (n = 3 independent samples). **b** Kaplan–Meier survival curves after injection of miR199a-5p mimics compared to injection of control mimic after MI injury, p value was generated by Log-rank (Mantel-Cox) test, n = 39 rats, mimic control; n = 31 rats, miR199a-5p mimics. **c** The echocardiographic images and echocardiography analyses of cardiac function after miR199a-5p mimic injection at indicated time points post-MI, (1 weeks: n = 30 rats, mimic control, n = 27 rats, miR199a-5p mimics; 4 weeks: n = 23 rats, mimic control, n = 27 rats, miR199a-5p mimics; 2 months: n = 22 rats, mimic control, n = 25 rats, miR199a-5p mimics; 4 months: n = 20 rats, mimic control, n = 25 rats, miR199a-5p mimics; 6 months: n = 11 rats, mimic control, n = 15 rats, miR199a-5p mimics; 8 months: n = 9 rats, mimic control, n = 10 rats, miR199a-5p mimics; 12 months: n = 8 rats, mimic control, n = 10 rats, miR199a-5p mimics). **d**–**k** Heart weight/tibial

length (HW/TL) ratio and lung weight/body weight (LW/BW) ratio of rats after miR199a-5p mimic injection at 6 months (**d**, n = 6 rats, mimic control; n = 9 rats, miR199a-5p mimics) and 12 months (**e**, n = 8 rats, mimic control; n = 10 rats, miR199a-5p mimics). LV internal dimension at end-systole (LVID's) at 6 months (**f**, n = 15 rats, mimic control; n = 20 rats, miR199a-5p mimics) and 12 months (**g**, n = 8 rats, mimic control; n = 10 rats, miR199a-5p mimics) post-MI, n = 6-10. Elisa detection of expression of pathological remodeling marker protein BNP in serum after miR199a-5p mimic injection at 6 months (**h**, n = 6 rats, mimic control; n = 9 rats, miR199a-5p mimics) and 12 months (**i**, n = 8 rats, mimic control; n = 10 rats, miR199a-5p mimics). Masson's staining displayed the fibrous tissue (blue) and myocardium (red) of sections of hearts from rats at 6 months (**j**, n = 6 rats, mimic control; n = 9 rats, miR199a-5p mimics) and 12 months (**k**, n = 8 rats, mimic control; n = 10 rats, miR199a-5p mimics) post-MI. Scale bars: 1 mm. Statistical analysis of infarct size of the infarcted heart. Data for (**c**)−(**k**) are presented as mean ± SD. The p-values were generated using an unpaired two-tailed Student's t-test. *P < 0.05, **P < 0.01, NS, not significant. Source data are provided as a Source Data file.

indicate that a single injection of miR199a-5p mimics during LAD ligation can provide long-lasting protection to cardiac function for up to 12 months after myocardial infarction.

### MARK4 and AGTR1 are co-targets of miR199a-5p

Subsequently, we conducted RNA sequencing to analyze the transcriptome of cardiomyocytes transfected with either miR199a-5p or a negative control (NC) mimic. The volcano plot showed differential genes and the top 10 genes were selected as candidate targets for further validation (Fig. 3a). We tested whether miR199a-5p mimics

could inhibit the expression of these candidate genes and miR199a-5p inhibitors could increase the expression of these candidate genes in NRVMs. As measured by qPCR, miR199a-5p mimics significantly inhibited agtr1 and itg8 mRNA expression compared with NC mimics, and miR199a-5p inhibitors only enhanced agtr1 mRNA expression (Fig. 3b). The study found that AGTR1, as an important component of the renin-angiotensin system, exhibits moderate expression in muscle cells of the vasculature, cardiomyocytes, and proximal tubules[25]. Figure 3c visualize the location of the predicted miR199a-5p recognition sites within the 3' untranslated region (3'UTR) of AGTR1 transcripts.

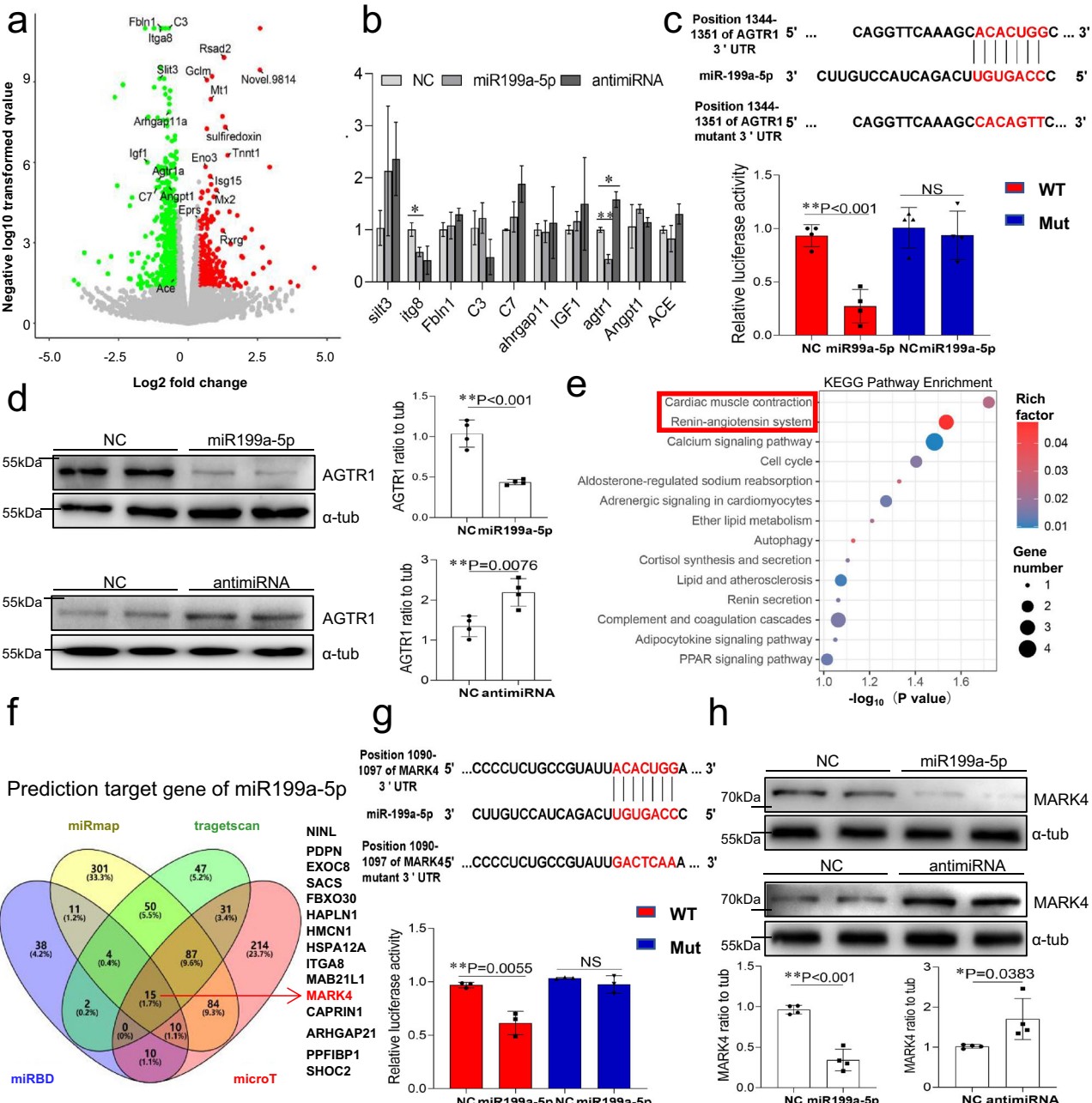

**Fig. 3 | MARK4 and AGTR1 are co-targets of miR199a-5p. a** The volcano plot displaying the differential expressed genes in cardiomyocytes transfected with miR199a-5p or NC mimic. Red and green colors indicate up-regulated or down-regulated genes. **b** qPCR of top 10 candidate target genes in miR199a-5p, NC mimic or antimiRNA treated NRVMs in vitro, n = 3 independent sample. **c** Luciferase vectors containing the miR199a-5p-binding sites of *AGTR1* were delivered to 293 T cells with or without co-delivery of miR199a-5p. Twenty-four hours after the transfection, luciferase activity was measured, n = 4 independent samples. **d** The cardiomyocytes were transfected with either miR199a-5p or antimiR199a-5p for 72 h. The expression levels of AGTR1 were examined by western blotting, n = 4 independent samples. **e** Bubble chart showing the enrichment result of biological process between miR199a-5p or negative control (NC) mimic transfected cardiomyocytes analyzed by KEGG pathway. **f** Predictive analysis of miR199a-5p key target

genes by miRBD, miRmap, tragetscan, and microT site. The number in the Venn diagram indicates the number of miRNAs belonging to the corresponding intersection. **g** Luciferase assays confirmed that miR199a-5p is associated with the MARK4 mRNA 3′-UTR. Constructs (vectors) carrying or not carrying the MARK4 3′-UTR were cotransfected with scramble NC mimics or miR199a-5p mimics in 293 T cells, n = 3 independent sample. **h** Western blot analysis of MARK4 protein expression in cardiomyocytes treated with NC, miR199a-5p or antimiRNA, n = 4 independent samples. Quantitative data were expressed as the mean ± SD. of at least 3 independent experiments. The p-values for Fig. 2d and h were generated using an unpaired two-tailed Student's t-test, p value for Fig. 2b, c, g were generated by one-way analysis of variance (ANOVA), followed by Tukey's multiple-comparison post hoc test. *P < 0.05, **P < 0.01, NS, not significant. Source data are provided as a Source Data file.

Luciferase reporter gene analysis demonstrated that miR199a-5p binds to the 3′UTR of AGTR1, resulting in their inhibition. The loss of inhibitory effect was observed when the binding sites were mutated. Furthermore, Western blot analysis confirmed that treatment with the miR199a-5p mimic reduced the protein expression of AGTR1 in

cardiomyocytes, while treatment with the miR199a-5p inhibitor increased their expression (Fig. 3d).

KEGG pathway analysis determined that miR199a-5p participates in the renin-angiotensin system and myocardial contractile pathway (Fig. 3e). To identify the target molecules of miR199a-5p in regulating

the myocardial contraction pathway, we utilized predictive systems (http://www.targetscan.org, http://mirdb.org, http://mirmap.ezlab.org, http://www.microrna.gr/microt_webserver/) to determine 15 potential target genes of miR199a-5p. Among them, MARK4 had a high binding score and species evolutionary conservation score (Context++ score percentile 97, PcT 0.81), and it regulated myocardial contractility without affecting infarct size or cardiac remodeling[26] (Fig. 3f). Therefore, we hypothesized that MARK4 might be a target of miR199a-5p involved in myocardial contraction. Luciferase reporter gene analysis showed that miR199a-5p bound to the 3′UTR of MARK4 (Fig. 3g), and western blotting analysis confirmed that miR199a-5p decreased MARK4 protein expression (Fig. 3h). These findings strongly suggested that MARK4 and AGTR1 are common target genes of miR199a-5p.

### miR199a-5p mitigates ROS and oxidative injury via AGTR1 in vitro

Given the ability of miR199a-5p to provide early cardiac protection immediately after MI, we sought to investigate its potential to inhibit MI-induced apoptosis. Neonatal rat ventricular myocytes (NRVMs) and H9C2 cells were transfected with miR199a-5p, inhibitors, or NC mimics to modulate the expression of miR199a-5p. Overexpression of miR199a-5p significantly enhanced the survival of AngII-treated NRVMs and H9C2 cells at 48 hours post-transfection, while inhibition of miR199a-5p reduced their survival (Fig. 4a, and S2a, b, c). Flow cytometry analysis revealed that the inhibitor group demonstrated an increase in apoptosis, while the miR199a-5p group demonstrated a decrease in apoptosis relative to the control group after 48 hours of AngII treatment (Fig. 4b).

We previously identified AGTR1 as a direct target of miR199a-5p and demonstrated that miR199a-5p inhibits AGTR1 mRNA and protein expression. It is known that AngII binding to AT1Rs stimulates NOX4 activity, leading to ROS production[27]. To determine whether miR199a-5P has an antioxidant effect during cardiomyocyte apoptosis, we transfected cardiomyocytes with miR199a-5p, inhibitors, or NC mimic, followed by detection of mitochondrial functional indexes after inducing apoptosis with AngII. As shown in Figure S2d, transfection of miR199a-5p in apoptotic cardiomyocytes attenuated intracellular and mitochondrial calcium overload, whereas inhibition of miR199a-5p led them to be exacerbated. We used flow cytometry to quantify cellular oxidative stress by 2′,7′-dichlorodihydrofluorescein (DCFH-DA) assay and found that inhibition of miR199a-5p increased oxidative stress in NRVMs (Fig. 4c) and in H9C2 cells (Fig. S2e). Western blot analysis revealed that overexpression of miR199a-5p significantly reduced NOX4 protein levels, while inhibition of miR199a-5p elevated NOX4 expression in NRVMs (Fig. 4d). We then investigated the effects of miR199a-5p on NRVM ROS generation, mitochondrial membrane potential (MMP) maintenance, and mitochondrial permeability transition pore (mPTP) opening. ROS levels were detected by fluorescence microscopy with DCFH-DA staining, changes in JC-1 fluorescence were used to record mitochondrial membrane potential, and the mPTP open state was determined by the calcineurin-AM/CoCl₂ assay. The results suggest that AngII increased ROS production, induced mPTP opening, and reduced mitochondrial membrane potential, which could be reversed by transfection of miR199a-5p (Fig. 4e). ROS is known to cause extensive damage to nucleic acids[28]. Therefore, we quantified oxidative modification of DNA bases by measuring 8-OHG. Immunofluorescence assay revealed that overexpression of miR199a-5p significantly reduced nuclear 8-OHG levels after NRVMs exposure of AngII, whereas inhibition of miR199a-5p increased nuclear 8-OHG levels (Fig. 4e). These findings suggest that miR199a-5p reduces ROS generation and protects NRVMs from oxidative damages.

To further elucidate whether the regulatory effect of miR199a-5p on oxidative injury is related to the suppressions of AGTR1, we treated NRVMs and H9C2 cells with AGTR1-specific inhibitor, irbesartan (IRB),

prior to AngII stimulation. Figures 4f–h and S2f–g demonstrate that IRB processing reversed the suppressive effects of miR199a-5p on NOX4 protein expression, ROS generation, mitochondrial membrane potential reduction, mPTP opening, DNA oxidative damages, and NRVMs apoptosis. Collectively, these results suggest that miR199a-5p attenuates ROS generation and oxidative damages through target AGTR1.

### miR199a-5p enhances cardiomyocyte contractility through microtubule detyrosination

Previous studies have highlighted the role of miR199a-5p in myocardial contraction. Therefore, we aimed to investigate whether increased miR199a-5p levels could enhance contractile function in cardiomyocytes during the pathogenic phase of cell injury. To assess the protective effect of miR199a-5p on doxorubicin-treated NRVMs, we performed cardiomyocyte contraction assays using the RTCA Cardiac System (ACEA Biosciences). The results demonstrated that treatment with 1 μM doxorubicin led to a decrease in the beating rate and amplitude of NRVMs compared to the control baseline. Pre-treatment with miR199a-5p mimic preserved the decrease in beating rate and amplitude, while inhibition of miR199a-5p exacerbated the decrease in beating rate and amplitude (Fig. 5a).

MARK4 has been reported to control microtubule detyrosination through MAP4 phosphorylation and promote VASH2 entry into microtubules[26]. Previous luciferase reporter analysis and western blotting suggest that miR199a-5p functions directly on MARK4 and reduces MARK4 protein expression. It is well documented that microtubules provide mechanical resistance to myocytes through interactions with sarcomeres and that these interactions are mediated by detyrosination-dependent binding to desmin, which regulates myocyte stiffness and contractility, and that excessive detyrosination increases contractile resistance[8]. In addition, inhibition of destyrosinated microtubules improves myocyte function in patients with myocardial ischemia and heart failure[9]. Based on these findings, we propose that miR199a-5p may modulate microtubule detyrosination by attenuating MARK4 expression to maintain myocardial contraction. We confirmed the decrease in detyrosinated microtubule proteins after transfection with miR199a-5p using a western blot assay, (Fig. 5b). Furthermore, we used siRNA to silence MARK4 and examined the impact of miR199a-5p on the spontaneous contractile function of damaged cardiomyocytes. The results demonstrated that the miR199a-5p-mediated control of detyrosinated microtubule proteins and regulation of contractility in cardiomyocytes was lost after MARK4 silencing (Figs. 5c, d, and S3a). Similarly, the regulatory effect of miR199a-5p on myocardial contractility was lost after silencing VASH2 using siRNA (Fig. S3b–d). The results suggest that miR199a-5p regulates cardiomyocyte contractility through the MARK4/VASH2/dTyr-tublin pathway.

### Generation and characteristics of the P-MSN/miR199a-5p delivery system

Previous results have shown that miR199a-5p protects cardiac function after MI in the long term, but the translational application of miRNAs is limited by the challenge of locally injecting damaged myocardium. To address this limitation, we developed nanoparticles that can be intravenously injected and target ischemic myocardium for MI treatment. The construction strategy of P-MSNs/miR199a-5p delivery nanocarriers is depicted in Fig. 6a. Initially, a positively charged amino-functionalized MSN (NH₂-MSN) was synthesized to link the peptide CSTSMLKAC through an amino and carboxyl group reaction. Subsequently, PEI was added to the solution to prepare PEI-modified peptide-MSNs complexes, enabling the adsorption of negatively charged miR199a-5p to form P-MSN/miR199a-5p nanoparticles (Fig. 6b). Additionally, the zeta potential values for pure miRNA, as well as for 1:5 and 1:10 miRNA:P-MSN complexation ratios, were determined to be 22.04 mV, 5.957 mV, and 10.43 mV, respectively (Fig. S4a).

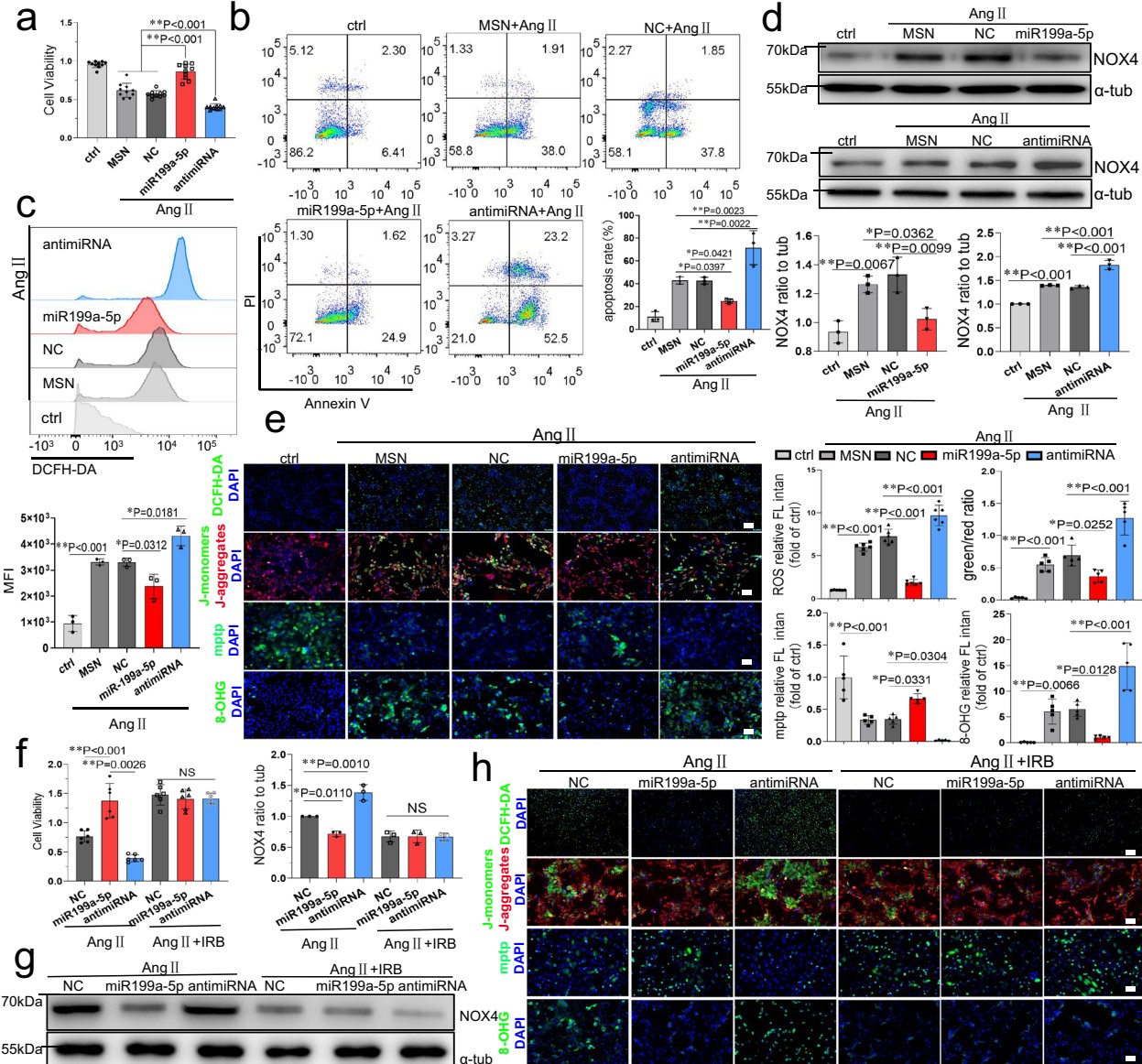

**Fig. 4 | miR199a-5p attenuates ROS and oxidative injury via AGTR1 in NRVMs.**
**a–e** The cardiomyocytes were transduced with miR199a-5p or antimiR199a-5p for 1 h and then incubated with/without Ang II(0.1 μM) for 48 hours. Supernatants were collected and cell viability was detected by CCK-8, n = 10 independent samples (**a**). Ang II-induced cell apoptosis was analyzed by flow cytometry, n = 3 independent samples (**b**). Total intracellular ROS in DCFH-DA staining was detected by flow cytometry, n = 3 independent samples (**c**). The protein levels of NOX4 were detected by western blotting, n = 3 independent samples (**d**). Intracellular ROS levels, mitochondrial membrane potential (MMP), mitochondrial permeability transition pore (mPTP) openness, and oxidative DNA damage were observed by fluorescence microscopy, n = 6 independent samples (**e**). **f–h** After transfecting with miR199a-5p or antimiR199a-5p for 1 h, NRVMs were cultured with Ang II and with or without irbesartan (IRB) for 48 hours. Detection of cell viability by CCK-8,n = 6 independent samples (**f**). Western blotting to detect NOX4 protein levels, n = 3 independent samples (**g**). Detection of ROS levels, MMP, mPTP openness, and oxidative DNA damage in NRVMs (**h**). Quantitative data were expressed as the mean ± SD. of at least 3 independent experiments. The p value for Fig. 3a–g were generated by one-way analysis of variance (ANOVA), followed by Tukey's multiple-comparison post hoc test. *P < 0.05, **P < 0.01, NS, not significant. Source data are provided as a Source Data file.

Further analysis was performed to evaluate the loading efficiency, release kinetics and miRNA protection of the nanoparticles. According to Fig. 6b, the optimal loading capacity of miRNA reached approximately 16.6 wt% (MSN: miR199a-5p = 5:1), and subsequent experiments were performed using the 16.6 wt% miR199a-5p/MSN complex (left and middle panels). The right panels demonstrated that the delivery vehicle effectively protected the packaged miRNAs from enzymatic degradation. Sustained release of the nanoparticles was observed, with an 80% launch efficiency within 3 days (Fig. S4b).

Subsequently, the cell viability of NRVMs cultured with P-MSN/miR199a-5p was assessed through cell live-dead staining and a CCK-8 assay, which revealed good cell viability after the addition of

nanoparticles in concentrations ranging from 0 to 80ug/ml concentration for 72 hours of culture (Fig. S4c, d). The ability of the nanoparticles to escape from endosomes was assessed, as miRNAs function in the cytoplasm and need to overcome endocytosis, blockage, and degradation by lysosomes[29,30]. The miRNA molecules were labeled with cy3 and the endolysosomes were stained with Lysotracker Green. The appearance of yellow dots in the merged images indicated the release of most miRNA molecules from the endolysosomes after 6 hours (Fig. 6c). The integrity of miR-199a-5p after lysosomal escape was evaluated by analyzing its expression in primary cardiomyocytes following the addition of P-MSN/miR-199a-5p nanoparticles. RT-qPCR results revealed a progressive rise in miR-199a-5p expression at 1, 3, 6,

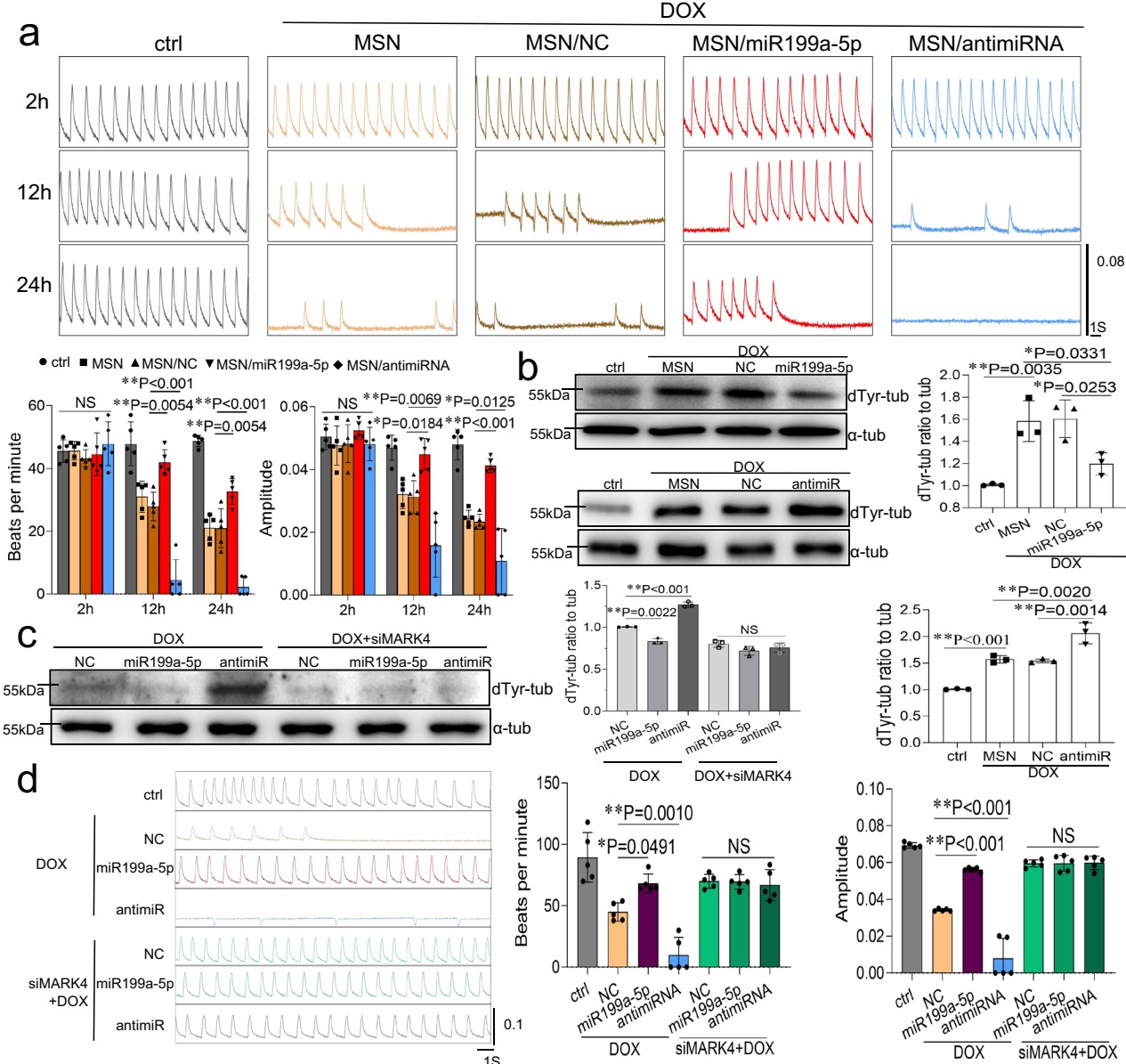

**Fig. 5 | miR199a-5p increases cardiomyocyte contraction by microtubule detyrosination. a** The cardiomyocytes were transduced with miR199a-5p or antimiR199a-5p for 1 h and then incubated with/without DOX(1 μM) within 24 hours. Label-free impedance analysis of spontaneously contracting NRVMs under different treatments using the RTCA cardio system. Representative images of NRVMs' real-time beating activity and quantification of its beating frequency and amplitude, n = 5 independent samples. **b** The protein level of dTyr-tub in NRVMs was detected by immunoblotting, n = 3 independent samples. **c** After transfecting with MARK4-targeting siRNAs 3# or NC siRNA for 48 hours, NRVMs were

transduced with miR199a-5p or antimiR199a-5p for 1 h and then incubated with DOX(1 μM) within 24 hours. The protein levels of dTyr-tub were detected by western blotting, n = 3 independent samples. **d** The transient pulse mode of NRVMs under different processing, n = 5 independent samples. Quantitative data were expressed as the mean ± SD. of at least 3 independent experiments. The p value for (**a**)–(**d**) were generated by one-way analysis of variance (ANOVA), followed by Tukey's multiple-comparison post hoc test. *P < 0.05, **P < 0.01, NS, not significant. Source data are provided as a Source Data file.

and 12 hours post-transfection (Fig. S4e). We then examined the cell transfection efficiency of P-MSN/miRNA-cy3. Flow cytometric analysis revealed that approximately 90.7% of NRVMs (Fig. 6d) and 96.2% of H9C2 cells (Fig. S4f) expressed cy3 fluorescence after 72 hours of transfection. Immunoblotting showed that P-MSN/miR199a-5p nanoparticles significantly reduced MARK4 protein expression compared to the controls (Fig. S4g). And P-MSN/miR199a-5p nanoparticles were transfected efficiently in NRVM and H9C2 cells(Fig. S4h). The persistence of the delivery system in cardiomyocytes was assessed by transfecting P-MSN/miR-cy3 and P-MSN/miR199a-5p nanoparticles into NRVMs. Intracellular cy3 fluorescence and miR-199a-5p levels were monitored over a 4-week period. The results showed sustained

cy3 fluorescence in cardiomyocytes throughout the entire 4 weeks, with peak miR-199a-5p levels detected after 1 week and maintained for over 4 weeks (Fig. S4i, j). P-MSN/miR199a-5p nanoparticles exhibited a substantial initial release of miRNA within the first week, followed by a slow and continuous release lasting up to 4 weeks. This release profile effectively aligned with the reduced fluctuation of miR199a-5p post-infarction, thereby improving the efficacy of P-MSN/miR199a-5p nanoparticles in infarct repair.

To investigate the targeting efficiency of P-MSN/miRNA nano-particles towards injured NRVMs, NRVMs were pretreated with $H_2O_2$ or left untreated, and then incubated with MSN/miRNA-cy3 or P-MSN/miRNA-cy3 for 6 hours. Flow cytometry analysis showed that P-MSN/

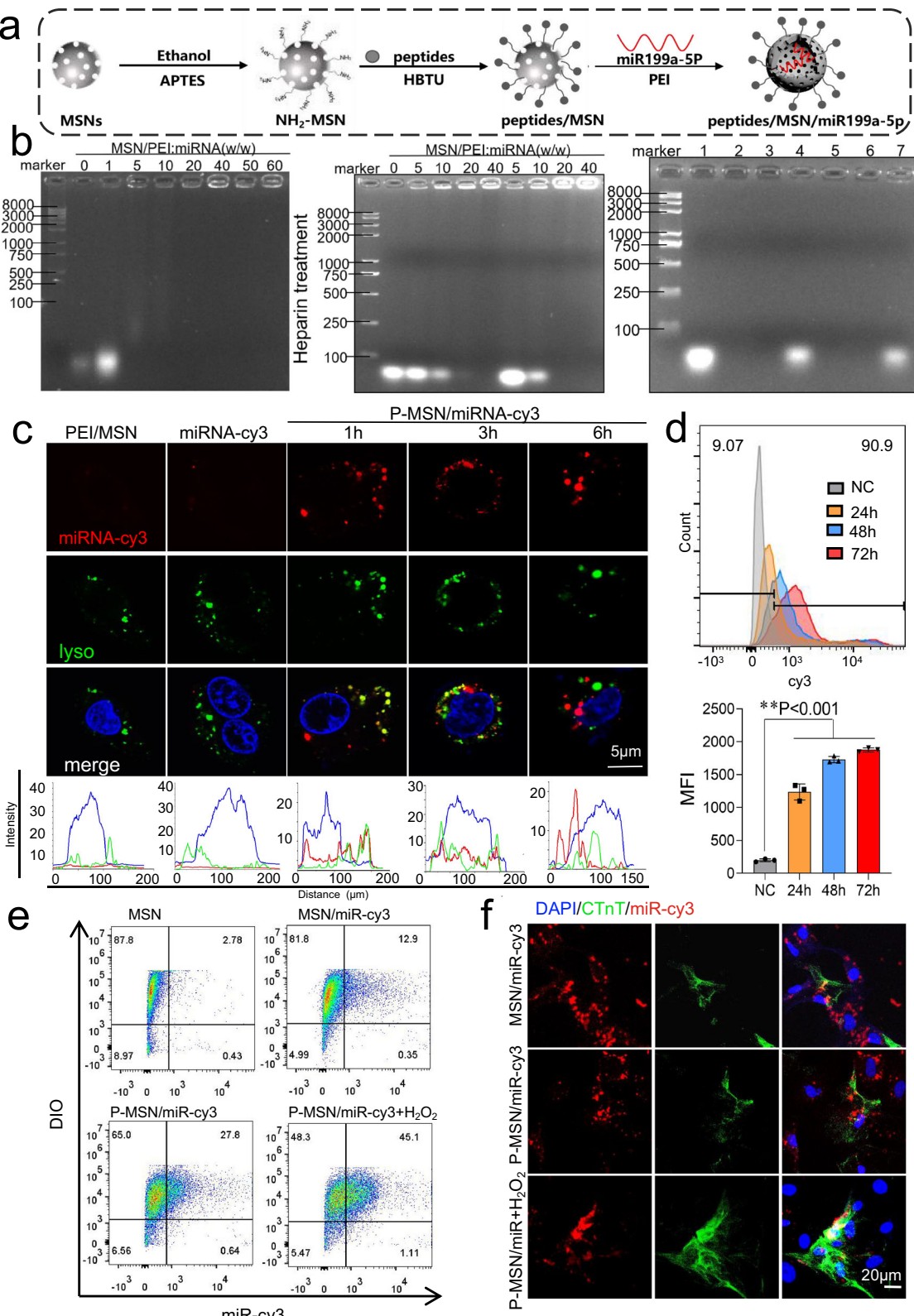

miRNA-cy3 bound to injured NRVMs more efficiently compared to the MSN/miRNA-cy3 group (Fig. 6e). And, primary cardiomyocytes were treated with or without $H_2O_2$ and incubated with P-MSN/NC, MSN/miR199a-5p, or P-MSN/miR199a-5p for 12 hours. The RT-qPCR analysis revealed no significant difference in the intracellular miR199a-5p expression between the MSN/miR199a-5p and P-MSN/miR199a-5p groups under normal conditions. However, following cell injury induced by $H_2O_2$, miR199a-5p expression was enhanced 1-fold in the P-MSN/miR199a-5p group compared to the MSN/miR199a-5p group (Fig. S4k). Additionally, co-culturing of cardiomyocytes and non-cardiomyocytes, with or without $H_2O_2$ pretreatment, followed by the addition of MSN/miRNA-cy3 or P-MSN/miRNA-cy3 for 6 hours, revealed that P-MSN/miRNA-cy3 predominantly localized to damaged cardiomyocytes rather than damaged non-cardiomyocytes (Fig. 6f).

**Fig. 6 | Construction of a nanoparticle targeting damaged cardiomyocytes loaded with miR199a-5p. a** Schematic construction of PEP peptide-modified MSN nanoparticles loaded with miR199a-5p. **b** The optimal loading capacity of miR199a-5p for the system was 5:1 (MSN: miR199a-5p), determined by gel retardation assay (left and middle panel)). Agarose gel electrophoretic analysis for evaluating the miRNA protective capability of P-MSN/miR199a-5p nanocarriers. Lane 1: Naked miR199a-5p; Lane 2: Naked miR199a-5p treated by RNase A; Lane 3: P-MSN/miR199a-5p; Lane 4: P-MSN/miR199a-5p incubated with heparin; Lane 5: P-MSN/miR199a-5p incubated with heparin, followed by treated with RNase A; Lane 6: P-MSN/miR199a-5p incubated with RNase A; Lane 7: P-MSN/miR199a-5p treated with RNase A first, and separated from the mixture subsequently; thereafter, such nanocarriers were incubated with heparin (right panel). **c** Fluorescent confocal laser scanning microscopic images of NRVMs after incubation with P-MSN/ miR199a-5p nanocarriers within 6 h, red fluorescent images of cy3-labeled miR199a-5p; green fluorescent images of endolysosomes stained by Lysotracker; bule fluorescent images of the nucleus, Bar= 5 μm. **d** Representative flow cytometry analysis of cy3- miR199a-5p transfection in NRVMs after 72 h of incubation with P-MSN/miR199a-5p nanocarriers (data are presented as mean ± SD, **P < 0.01, p value was generated by one-way analysis of variance (ANOVA), followed by Tukey's multiple-comparison post hoc tes, n = 3 independent samples). **e** Flow cytometry analysis of cy3-labeled MSN/miR199a-5p or P-MSN/miR199a-5p binding to $H_2O_2$- treated or untreated NRVMs labeled with DIO. The percentage of Q2 indicates the proportion of NRVMs internalizing cy3-labeled miR199a-5p, n = 3. **f** Confocal microscopy images of colocalization of cy3-labeled miR199a-5p(red) and the FITC-labeled CTnT (green) in NRVMs treatment with or without $H_2O_2$. Cell nuclei were dyed with DAPI (blue). Bar=20μm. Source data are provided as a Source Data file.

This demonstrated the targeting ability of P-MSN/miRNA nanoparticles on injured NRVMs.

## Therapeutic potential of P-MSN/miR199a-5p nanoparticle in infarcted hearts

After establishing the crucial role of miR199a-5p in cardiac protection following cardiac injury, our next inquiry was whether P-MSN/ miR199a-5p complexes could be employed as a potential therapeutic approach for the treatment of MI. The experimental scheme is shown in Fig. 7a. As shown in Fig. 7b, fluorescence imaging of rats was performed at 2, 6, 24, and 48 hours after nanoparticle injection. The fluorescence in the chest and abdomen of the rats gradually increased after injection, and the fluorescence signal in the chest was stable and clear after 6 hours, peaked after 24 hours and kept until 48 hours. Statistical analysis showed that the fluorescence signal detected in the breast of P-MSN/miR199a-5p-treated infarcted rats was significantly stronger than that of MSN/miR199a-5p-treated infarcted rats and P-MSN/miR199a-5p-treated control rats. Following a 4-week tail vein injection of P-MSN/miR199a-5p in infarcted rats, miR199a-5p expression was detected in both the remote and infarct zones. RT-qPCR results indicated a notable increase in miR199a-5p expression in the infarcted region, with up to a 15-fold increase at 7 days post-injection maintained at a 5-fold increase by the 4th week (Fig. S5a). FISH analysis of heart tissues confirmed increased miR199a-5p expression in the infarct region of rats treated with P-MSN/miR199a-5p compared to controls (Fig. S5b). Furthermore, analysis of target protein expression in vivo revealed that P-MSN/miR199a-5p led to a reduction in the levels of MARK4, AGTR1, as well as downstream pathway proteins such as dTyr-tub and NOX4 (Fig. 7c).

Considering the specific role of miR199a-5p in inhibiting cardiomyocyte apoptosis and enhancing cardiomyocyte contractility in vitro, we further evaluated the therapeutic efficacy of P-MSN/miR199a-5p for MI in vivo. Apoptosis in infarcted hearts was assessed using a TUNEL assay, 7 days after tail vein injection of P-MSN/miR199a-5p. The results revealed a substantial decrease in TUNEL signal in the hearts injected with P-MSN/miR199a-5p. (Fig. 7d). Infarcted myocardial contractility was then measured during electric field cycling stimulation, and Fig. 7e suggests that compared with negative controls, infarcted myocardial contractility was significantly increased after overexpression of miR199a-5p, and contractility was increased more by injection of P-MSN/miR199a-5p than by injection of MSN/miR199a-5p. Echocardiography results revealed that injection of P-MSN/miR199a-5p and injection of MSN/miR199a-5p significantly restored cardiac function after MI, and the effect of P-MSN/miR199a-5p was preferred to that of MSN/miR199a-5p (Fig. 7f, detailed data are shown in Supplementary Date 2). Reductions in heart and lung weights were observed in rats treated with P-MSN/miR199a-5p, with superior outcomes compared to MSN/miR199a-5p (Fig. S5c). Histological examination by Masson's trichrome staining revealed that overexpression of miR199a-5p significantly reduced the size of the fibrotic area of the heart post-MI compared with the negative control, and P-MSN/miR199a-5p had the

superior outcome (Fig. 7g). Accordingly, RT-qPCR analysis confirmed that miR199a-5p reduced the expression of the cardiomyopathy marker genes nppa and Mhy7 as well as the genes expression of the collagen marker Col1a1, Col1a2, and Col3a1 (Fig. S5d). H&E staining experiments were performed on the infarcted area of rats in each group at four weeks post-infarction. The results showed that the P-MSN/miR199a-5p group had the least amount of scar tissue and the most amount of myocardial tissue, indicating the optimal repair effect (Fig. S5e). In conclusion, P-MSN/miR199a-5p nanoparticles target ischemic myocardial tissues, releasing miR199a-5p targeting AGTR1 to reduce oxidative damage in the early post-infarction period of myocardial infarction; and targeting MARK4 to affect the long-term myocardial contractility, thus achieving long-term infarction repair effects (Fig. 8).

## Discussion

In this study, we investigated the role of a specific microRNA, miR199a-5p, in protecting the heart against MI. Through both in vivo and in vitro experiments, we discovered that miR199a-5p reduced fibrosis area and modified cardiac function after MI by enhancing myocardial contractility and inhibiting cardiomyocyte apoptosis. Additionally, we developed a nanoparticle-based delivery system for efficiently delivering miR199a-5p to the ischemic region of the heart. When administered to rats after MI, this system demonstrated significant cardioprotective effects.

One of the key findings of this study is that miR199a-5p appears to shield the heart from MI damage through targeting two different biological processes at different phases. In the early stages of MI, miR199a-5p reduced cardiomyocyte apoptosis and improved cardiac function via targeting AGTR1. In the late stages, when cardiac contractile function was weakened, miR199a-5p enhanced myocardial contractility to mitigate post-infarction heart failure via targeting MARK4. This dual mechanism of miR199a-5p-mediated cardioprotection presents promising opportunities for the development of effective treatments for MI and heart failure.

It is generally accepted that miRNAs fulfill their functions by targeting many downstream mRNA targets[13]. We also investigated the downstream mRNA targets of miR199a-5p and discovered that it suppressed the expression of AGTR1 and MARK4 proteins. AGTR1 is known to increase oxygen consumption and superoxide production via Nox4. This effect was confirmed in mitochondria of AT1 receptor-deficient mice with decreased respiration rates[31,32]. The Nox4 isoform of the NADPH oxidase family members is localized within the cell membrane of cardiomyocytes[33] and renal cells[34]. Many studies have shown that there is an interaction (i.e., crosstalk signaling) between the membrane-bound NADPH oxidase complex and mitochondria that is ROS-mediated, and thus NADPH oxidase-generated ROS may act as triggers to evoke the opening of the ATP-sensitive potassium channel (mitoKATP), which leads to the mitochondrial generation of ROS[35]. Here, we verified that AGTR1 is directly involved in miR199a-5p-mediated regulation of cardiomyocyte apoptosis. We found that

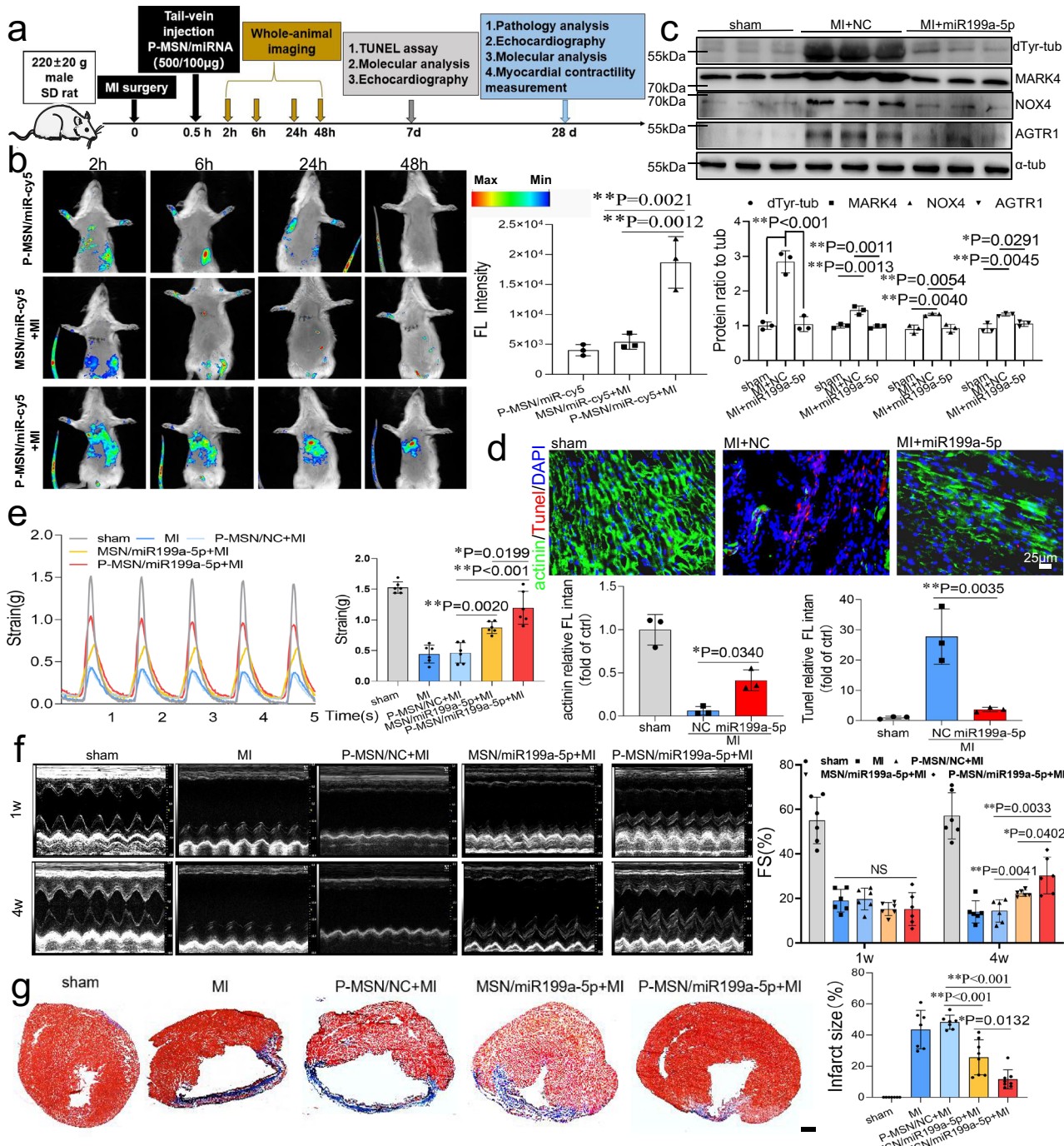

**Fig. 7 | P-MSN/miR199a-5p reduces infarct size and cell death after myocardial infarction and improves myocardial contractile function. a** Experimental design. **b** Rats underwent with or without MI injury and received cy5-labeled MSN/miR199a-5p or P-MSN/miR199a-5p treatment through tail intravenous injection. Fluorescence signals were detected within 48 hours after injection, n = 3 rats. **c** Expression of dTyr-tub, MARK4, NOX4, and AGTR1 proteins in cardiac tissues 7 days after infarction were detected by immunoblotting, n = 3 rats. **d** The sections of hearts from the sham-operated group and 7 days after myocardial infarction in NC or miR199a-5p mimic-treated rats were labeled with Tunel staining (red) and immunolabeled with α-actinin (green) and nuclei (blue), n = 3 rats. Bar=25μm. **e** Comparison of representative contraction traces in the sham group, the MI group, the P-MSN/NC + MI group, the MSN/miR199a-5p+MI group, and the P-MSN/

miR199a-5p+MI group for 5 seconds. Statistical analysis of maximum strain in different groups of injured hearts, n = 5 rats. **f** The echocardiographic images of 1 week (above) and 4 weeks(bottom) post-MI injury in the sham group (control group), the MI group, P-MSN/NC + MI, MSN/miR199a-5p+MI and P-MSN/miR199a-5p+MI group. Representative parameters of left ventricular function based on echocardiography of each group at 4 weeks after MI injury, n = 6 rats. **g** Mason's staining displayed the fibrous tissue (blue) and myocardium (red) of sections of hearts from rats in different groups. Scale bars: 1 mm. Statistical analysis of infarct size of the infarcted heart in different groups, n = 6 rats. Data for (**b**)–(**g**) are presented as mean ± SD. The p value for (**b**)–(**g**) were generated by one-way analysis of variance (ANOVA), followed by Tukey's multiple-comparison post hoc test. *P < 0.05, **P < 0.01, NS, not significant. Source data are provided as a Source Data file.

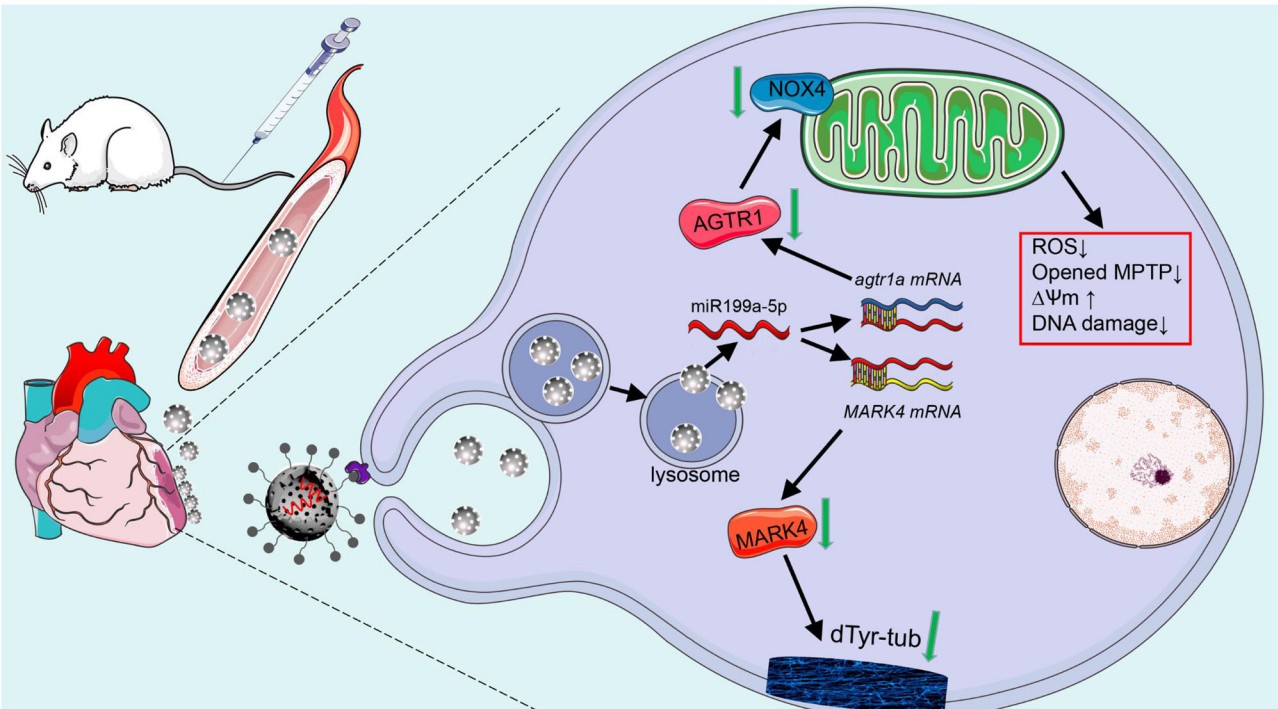

**Fig. 8 | Schematic representation of the main findings of this study.** PEP peptide-modified MSN nanoparticles loaded with miR199a-5p were injected tail-vein into myocardial infarction rats and targeted to ischemic myocardial tissue. Cardio-myocytes endocytosed the nanoparticles, released miR199a-5p to inhibit MARK4 and AGTR1 protein expression, and promoted cardiac repair after myocardial infarction by decreasing ROS anti-apoptosis and enhancing myocardial contractility.

miR199a-5p, mediated by inhibition of AGTR1, could decrease NOX4 protein expression and then lead to low ROS production and a significant increase of MMP for preventing mPTP opening. miR199a-5p overexpression could reduce oxidative DNA damage by scavenging NRVM mitochondrial ROS leading to reduced apoptosis. MARK4, another target gene of miR199a-5p expressed in the heart[36], is an evolutionarily conserved serine-threonine kinase[37,38] known to phosphorylate MAPs including tau, MAP2, and MAP4 at the KXGS motif within their microtubule-binding repeats[39]. MAP phosphorylation triggered by MARK induces a conformational change that alters MAP binding to microtubules, thereby regulating the detyrosination of microtubule proteins, which plays an important role in regulating myocardial contractility in the ischemic heart[26,40]. In this study, we demonstrated that miR199a-5p could also modulate cardiomyocyte contractility by influencing the detyrosination of microtubule proteins via regulating MARK4 in vivo and in vitro. The mechanistic studies confirmed that miR199a-5p could protect the heart from myocardial infarction injury by controlling the AGTR1/NOX4 and MARK4/VASH2/dTyr-tub pathways.

Current miRNA therapeutic strategies for MI face challenges in terms of low cellular uptake and poor tissue specificity[41]. In this study, we developed a p-MSN/miR199a-5p nanoparticle delivery system that efficiently transfects cells, protects miRNAs for long-term stability and targets ischemic myocardial tissue. The MSNs, as a delivery vector for miRNA, present good biocompatibility and high transfection efficiency[42,43]. In addition, it has been shown that MSNs show great potential for inhibiting post-inflammatory M1 polarization in vitro and in vivo[44]. In our study, we compared the use of degradable MSN-PEI and commercial transfection reagents (PEI, lipo3000, Ribo transfection regent) for miRNA intracellular delivery. Compared with these commercial transfection reagents, MSN-PEI is a more potent intracellular drug delivery vector with higher loading efficiency, long-term protection of miRNAs, and lysosomal escape ability. Cy3-miRNA uptake and Western blots studies demonstrated that MSN-PEI could efficiently transfect primary cardiomyocytes and H9C2 cells, and that the miR199a-5p-MSN-PEI complex had better inhibition efficiency than commercially available transfection reagents in vitro. And MSN-PEI protected miRNA from degradation by RNA enzymes within 21 days. To achieve tissue-specific miRNA delivery, we introduced the CSTSMLKAC peptide (PEP) on the surface of MSNs to target ischemic myocardium. In vitro experiments showed that p-MSN/miR199a-5p was more efficiently internalized by hydrogen peroxide-injured primary cardiomyocytes than MSN/miR199a-5p. Significantly enhanced ability of p-MSN/miR199a-5p targeting ischemic myocardium was observed by in vivo follow-up experiments. Furthermore, we demonstrated that p-MSN/miR199a-5p significantly inhibited cardiomyocyte apoptosis and enhanced myocardial contractility simultaneously. These synergistic effects led to reduced infarct size and improved cardiac function post-MI. In summary, the p-MSN/miR199a-5p delivery system can be taken as a promising nonviral nanodrug to achieve targeting treatment for damaged myocardium. Future applications in the therapy of clinical myocardial infarction can be envisioned.

The limitations of this study focus on the small animal model, the rat. The findings must be validated in a large animal model to further explicate the clinically meaningful targets and interventions revealed in the study. Furthermore, the cause of decreased miR199a-5p expression in MI and other cardiac diseases should be explored to provide a more comprehensive understanding. Further studies could start from the fact that large amounts of ROS released from mitochondria may result in decreased miR199a-5p synthesis or increased miR199a-5p degradation. Third, the slow degradation rate and long degradation time of silicon-based nanoparticles may lead to the accumulation of silicon-based materials in target organs. Exploring easily customizable alternative delivery vectors can facilitate smoother clinical translational applications in the future.

In conclusion, our findings suggest that miR199a-5p enhances myocardial contractility, inhibits cardiomyocyte apoptosis, and protects the heart from MI injury in the long term. Mechanistic studies confirmed

its role in controlling the AGTR1/NOX4 and MARK4/VASH2/dTyr-tubulin pathways. The study also constructed a p-MSN/miR199a-5p nanoparticle delivery system that efficiently transfects cells and targets ischemic myocardial tissue. In vivo results showed that p-MSN/miR199a-5p significantly inhibited cardiomyocyte apoptosis, enhanced myocardial contractility, and reduced infarct size. The results suggest that the p-MSN/miR199a-5p delivery system is expected to be an effective therapeutic strategy for MI and post-infarction heart failure.

## Methods

### Study approval
The procurement and utilization of these laboratory animals were carried out in strict accordance with the guidelines of the ethics committee and received approval from the Southern Medical University Animal Ethics Committee(SYXK(粵)2021-0167).

### Animal and reagents
Male Sprague-Dawley (SD) rats, weighing around 220 ± 20 g and with an age of 7-8 weeks, were obtained from the Laboratory Animal Center at the Academy of Southern Medical University. Small-interfering RNA (siRNA, siBDM1999A), control siRNA(siN0000001-1-10), miR199a-5p mimic(miR10000872-1-5), miR199a-5p inhibitor(miR2CM001) and control miRNA(miR01202-1-5) were synthesized by RiboBio Biotechnology (Guangzhou, China). miR199a-5p mimic 5′-3′:CCCAGU-GUUCAGACUACCUGUUC. 3′-5′:GAACAGGUAGUCUGAACACUGGG, miR199a-5p inhibitor 5′-3′: GAACAGGUAGUCUGAACACUGGG. Angiotensin II (AngII), $H_2O_2$, doxorubicin (DOX), irbesartan (IRB) and Polyethylenimine (PEI) were purchased from Sigma-Aldrich (St. Louis, MO, USA). Alexa Fluor-568 donkey anti-rabbit IgG (H&L) and Alexa Fluor-488 donkey anti-mouse IgG (H&L) were from Invitrogen. The Cell Counting Kit-8 (CCK-8) was provided by New Cell and Molecilar Biotech (china). Primary antibodies targeting Angiotensin II Type 1 Receptor (AGTR1, ab239995, WB use concentration:1 μg/ml), cardiac troponin T (CTNT, ab10214, IF use concentration:1 μg/ml), α-actinin (ab9465, IF use concentration:2 μg/ml), 8-OHG (ab62623, IF use concentration:2 μg/ml), alpha Tubulin (α-tub, ab7291,WB use concentration:1 μg/ml), NADPH oxidase 4 (NOX4, ab154244, WB use concentration:1 μg/ml), Detyrosinated alpha Tubulin (dTyr-tub, ab254154, WB use concentration:1 μg/ml) and MARK4 (Cell Signaling Technology, 4834, WB recommended concentration:1 μg/ml) were purchased from Abcam in Cambridge, UK.

### Data source from GEO data repository and differential expression analysis
GEO (http://www.ncbi.nlm.nih.gov/geo)[45] is a publicly available database that houses a vast collection of high-throughput sequencing and microarray datasets contributed by research organizations worldwide. In our study, we conducted a search within GEO using miRNA and cardiovascular disease as keywords to identify relevant gene expression datasets. Our inclusion criteria were set to select datasets with the largest sample size from the same sequencing platform with two independent expression profiles. Subsequently, we downloaded a total of eight microarray datasets from the database. To analyze these datasets, we utilized GEO2R (www.ncbi.nlm.nih.gov/geo/ge2r)[46], an online analysis tool that relies on two R packages, namely GEOquery and Limma. The GEOquery package facilitated the extraction and reading of the data, while the Limma package was employed to calculate the fold changes in differential gene expression. By employing GEO2R, we compared gene expression profiles between different groups to identify Differentially Expressed Genes (DEGs) associated with the disease condition and control groups. We removed probe sets without corresponding gene symbols and averaged genes with multiple probe sets. Only genes meeting the criteria of an adjusted P value < 0.05 and |logFC (fold change)| ≥ 1 were considered as DEGs. We obtained the common DEGs by utilizing an online Venn diagram tool.

### Synthesis of P-MSN/miRNA
The Mesoporous silica nanospheres (MSN) were procured from jkchemical (Beijing, China), while the PEP peptide (CSTSMLKAC) was synthesized by Taopu Biotech (Nanjing, China). The creation of microsphere nanoparticles involved dissolving 1 gram of microspheres in 100 milliliters of anhydrous toluene, followed by subjecting them to sonication. To this solution, 1 milliliter of APTS was added and allowed to incubate for 20 hours. After undergoing centrifugation, the particles were washed three times with ethanol and deionized water, and then lyophilized to obtain the MSN-NH$_2$ nanoparticles. Subsequently, 100 milligrams of MSN-NH$_2$ nanoparticles were dispersed in 20 milliliters of PBS, with the addition of 0.03 milligrams of n-n-hydroxysuccinimide (NHS) and 0.3 milligrams of benzotriazole-n, n, n, n, n, n, n-tetramethylureidoammonium hexafluorophosphate (HBTU) along with the PEP peptide. This mixture was incubated for 28 hours at 4 °C, followed by centrifugation and washing the resulting PEP-MSN nanoparticles three times at 8000 pcf for 10 minutes each 4 °C. The nanoparticles were then freeze-dried. The PEP-MSNs were dispersed in PEI aqueous solution and incubated at 4 °C for 20 h. The PEP-MSN-PEI nanoparticles were prepared by washing with deionized water for 3 times and then freeze-dried. Before use, P-MSN/miRNA nanoparticles were prepared by mixing PEP-MSN-PEI with miRNA in different mass ratios in PBS for 15 min.

### Characterization of P-MSN/miRNA
The size, zeta potential, and morphology of the P-MSN/miRNA nanoparticles were characterized using two techniques: dynamic light scattering (DLS) with a Malven Zetasizer Nano ZS90 and transmission electron microscopy (TEM) with an FEI, TecnaiG2F20.

### Cell culture and treatment
Primary cultured neonatal rat ventricular myocytes (NRVMs) were obtained from the hearts of S-D rats aged 1-3 days, utilizing well-established methods previously reported by our research group[47–49]. Rats were anesthetized with isoflurane, then their hearts were excised and removed. The vessels, atria, and pericardium were carefully transected and then rinsed twice with PBS. The ventricular myocardium was digested into a single-cell suspension using 0.25% trypsin and 0.1% type I collagenase. Suspended cells were centrifuged at 900 g for 5 minutes and then incubated on plates for 2 hours to remove noncardiomyocytes. To induce apoptosis, the cells were treated with 0.5 μM AngII for 72 hours or 1 μM DOX for 24 h. The cells were treated with 0.5 mM $H_2O_2$ for 8 h to induce oxidative damage. In some cases, activation of AGTR1 by AngII was inhibited using 10 μM IRB. H9C2(CL-0089) and HEK 293 T(CL-0005) cells were procured from Pricella Biotechnology (Wuhan, China).

### Intra-cardiac injection of MSN/miRNA
220 ± 20 g male rats were randomly assigned to receive intracardiac injections of MSN/miR199a-5p or MSN/control mimics (60 μg per rat heart) following myocardial infarction. Following the ligation of the left anterior descending (LAD) coronary artery, an insulin syringe was employed to inject a total volume of 50 μL of MSN/miRNA solution. The distribution of MSN/miRNA mimics was evenly performed in three regions surrounding the infarcted area, specifically the apex region, lateral wall, and anterior wall. An average volume of approximately 16 μL of mimic solution was injected into each of these regions. Careful attention was given to position the needle within the ventricular myocardial wall to prevent penetration into the ventricular cavity.

### Measurement of cardiac function by echocardiography
Left ventricular function was assessed in rats using an IE33 echocardiography system (Vevo2100, Visual Sonics) at various time points after the establishment of the MI model. The rats were anesthetized using isoflurane (induced with 3% isoflurane, then maintained with 2% of

isoflurane). M-mode and B-mode reflecting the morphology of the anterior wall of the left ventricle and beat-to-beat activity were recorded using the M250 sensor, respectively. Cardiac function indices, including Fractional Shortening (FS) and Ejection Fraction (EF), were determined by calculating values from three consecutive cardiac cycles.

## Histology and TUNEL assays

Rats were euthanized at different time intervals. The hearts were collected and sliced into frozen sections that were 6 μm thick. To determine the histologic characteristics of myocardial infarction, Masson trichrome staining was performed according to the instructions provided by the manufacturer. The infarct size was determined by calculating the ratio of collagen (blue) to myocardial muscle (red) in the frozen heart section. To assess tissue damage, 5 μm thick sections of the liver, lung, kidney, and spleen tissues were embedded in paraffin and mounted on polylysine glass slides. The slices were then H&E stained. Image J software was used to analyze the staining data.

To detect apoptotic cardiomyocytes, a Terminal deoxynucleotidyl transferase-mediated nick end labeling (TUNEL) assay was performed on the frozen sections. The One-Step TUNEL Apoptosis Detection Kit (Cat #C1089 Beyotime) was employed according to the instructions provided by the manufacturer. To identify the cardiomyocytes, α-actinin was utilized for staining.

## Measurement of BNP

At various time intervals following myocardial infarction, rat serum was collected. The levels of B-type natriuretic peptide (BNP) in the serum were measured using a commercially available ELISA kit obtained from eBioscience in San Diego, CA, USA.

## Quantitative RT-PCR

TRIzol Reagent (Invitrogen) was used to isolate total RNA from cellular or tissue samples. A 20 μL reaction system containing 2.0 μg of RNA sample was employed for reverse transcription to cDNA using random hexamers (Invitrogen). The obtained cDNA was subsequently utilized for quantitative reverse transcription-polymerase chain reaction (qRT-PCR) analysis to assess the expression of mRNA. For each analysis, 0.15 μL of the cDNA was employed. For miRNA expression analysis, the TaqMan® MicroRNA Reverse Transcription Kit (ABI) was used to reverse transcribe 10 ng of RNA sample into cDNA. A 1.5 μL cDNA pool, obtained from this reverse transcription, was used for quantitative PCR using the TaqMan® MicroRNA Detection Kit for each analysis. qPCR was performed using Applied Biosystems 7500 Real-Time PCR system. Using the $2^{-\Delta\Delta Ct}$ method, the relative expression of mRNA and miRNA relative to the reference genes GAPDH and U6 snRNA was determined. Primer sequences are shown in Supplementary Date 3.

## RNA-seq and transcriptome analysis

Three days following transfection with miR199a-5p or control mock, primary cardiomyocytes were subjected to RNA-seq analysis (three biological replicates per group). Novogene (Beijing, China) conducted the RNA-seq experiments. First, total RNA was extracted from the cell using TRIzol (Invitrogen). mRNA was subsequently pureed from total RNA with polyT oligo magnetic beads. Sequencing libraries were generated using the NEBNext® UltraTM RNA Library Preparation Kit for Illumina® (NEB, USA) according to the manufacturer's recommendations and index codes were added to the sequences for each sample. Samples with index codes were clustered on the cBot Cluster Generation System using the TruSeq PE Cluster Kit v3-cBotHS (Illumia) according to the manufacturer's instructions. After clustering was generated, library preparations were sequenced on the Illumina Hiseq platform and 150 bp paired-end reads were generated. For data analysis, the raw data in fastq format (raw reads) are first processed by an internal Perl script. Clean data (clean reads) were obtained by eliminating reads containing adapters, reads containing ploy-N and low

quality reads from the raw data. Reference genome and gene model annotation files were downloaded directly from the genome website. The reference genomes were indexed and aligned to pairs of clean reads using STAR (v2.5.1b) with the Maximum Mappable Prefix (MMP) method. HTSeq v0.6.0 was used to count the reads mapped to each gene. Differential expression analysis was carried out using the edgeR R package (version 3.12.1), with the P values adjusted using the Hochberg and Benjamini method. KEGG and GO pathway analysis was performed using the clusterProfiler R package. Hierarchical clustering heatmaps were created with the ggplot library.

## Western blot analysis

Protein samples obtained from primary cardiomyocytes or H9C2 cells were subjected to SDS-PAGE for separation. The separated proteins were then transferred onto polyvinylidene fluoride (PVDF) membranes (Millipore, Billerica, MA, USA). The membranes were blocked in TBS solution containing 0.05% Tween-20 and 5% BSA for 1 hour at room temperature. Subsequently, the membranes were incubated with primary antibodies at 4 °C overnight or 2 hours at room temperature. Following this, the membranes were incubated with corresponding secondary antibodies conjugated with horseradish peroxidase (HRP) for 1 hour at room temperature. Finally, the detection of the protein of interest was performed using enhanced chemiluminescence (Thermo Fisher, Carlsbad, CA, USA) according to the manufacturer's instructions. Data quantification was performed by image j v1.50 software.

## Luciferase reporter assay

For luciferase reporter gene assays, WT-AGTR1, mut-AGTR1, WT-MARK4 and mut-MARK4 dual luciferase plasmids were constructed by Genecefe Biotechnology Co (Wuxi, China). These plasmids were transfected into HEK293T cells using Lipofectamine 2000 (Invitrogen). For transfection, cells ($5\times10^4$) were seeded into 96-well plates and cultured overnight. Cells were then co-transfected with 80 ng of wild-type (WT) or mutant (Mut) dual-luciferase plasmid and miR199a-5p or control mimic (final concentration: 50 nM). 24 or 48 hours later, luciferase activity was assessed utilizing the Dual-Luciferase Reporter Assay System (Promega) following the guidelines provided by the manufacturer. The obtained data were normalized by dividing the firefly luciferase activity by the Renilla luciferase activity.

## Annexin V-FITC/PI staining

The apoptosis rate of NRVMs was detected using the Annexin V-FITC/PI Apoptosis Detection Kit. Briefly, NRVMs were suspended into 0.1 mL of binding buffer followed by a 15-minute incubation with Annexin V-FITC/PI buffer under light-free conditions. The fluorescence intensity was measured with the FACScan instrument from BD Biosciences, and the data were subsequently analyzed utilizing FlowJo 8.0 software.

## ROS detection

Detection of total cellular ROS was performed by DCFH-DA (Sigma-Aldrich). NRVMs at a density of $10^5$ cells/well were incubated in 10 mM DCFH-DA working solution for 30 min at 37 °C in the dark. Images were acquired with a fluorescence microscope (Olympus BX53 software). Fluorescence density was also detected by flow cytometry.

## MMP measurement

The Mitochondrial Membrane Potential Assay Kit utilizes JC-1 (Beyotime) to assess changes in mitochondrial membrane potential (MMP) following the manufacturer's guidelines. In summary, NRVMs were washed with PBS, followed by suspension of NRVMs in JC-1 staining solution and incubation at 37 °C for 20 min away from light. A fluorescence microscope (Olympus BX53 software) was employed to acquire the images. When the MMP in the cells is low, JC-1 remains monomeric and fluoresces green. In cells with high MMP, JC-1 forms a complex, the J-aggregate, which emits orange-red fluorescence.

The higher the ratio of red to green fluorescence, the more polarized the mitochondrial membrane is.

## Calcein-AM/CoCl₂ assays

The assessment of mitochondrial permeability transition pore (mPTP) opening was performed using the calcein-AM/CoCl₂ assay. Incubation of NRVMs with 5 mM calcein-AM at 37 °C for 30 min. Subsequently, they were incubated for another 20 min at 37 °C with 40 mM CoCl₂. In the absence of mPTP opening, the mitochondria retained calcein fluorescence even in the absence of CoCl₂. Conversely, when mPTP opening occurred, calcineurin within the mitochondria dissipated, resulting in the loss of calcein fluorescence. The fluorescence intensity was determined using fluorescence microscopy.

## Immunostaining

Immunofluorescence staining was performed on mouse α- actinin (dilution 1:250), mouse cTNT (dilution 1:250), and mouse 8-OHG (dilution 1:500) at 4°C overnight. Subsequently, they were treated with Alexa Fluor-488 or Fluor-568 fluorescent secondary antibody (both diluted to 1:500) for 1 hour. Subsequently, the stained sample was incubated with DAPI for 10 minutes to observe the nucleus. A fluorescence microscope (Olympus BX53 software) was employed to acquire the images.

## Label-free measurement of cardiomyocyte contraction

Real-time label-free impedance measurements (cellular index) of cardiomyocyte beating frequency (beats per minute) and contraction behavior were performed using the xCelligence RTCA cardio system (ACEA Biosciences). Impedance signals were monitored and recorded at a recording time of 20 seconds per scan. The basal contractile behavior of the cells was analyzed two days after culturing NRVMs into 96-well plates. Cardiomyocyte contraction was detected in real time by adding the corresponding drug and miRNA to the maintenance medium.

## Cellular uptake studies

NRVMs and H9C2 cells were cultured in Dulbecco's Modified Eagle's Medium (DMEM) supplemented with 15% fetal bovine serum and 1% penicillin-streptomycin. Cy3-miRNA (similar in size to siRNA) was used instead of miRNA for visualization to assess the extent of cellular uptake of the p-MSN/miRNA complex. Images were captured using fluorescence microscopy and flow cytometry was used to detect transfection efficiency.

To investigate whether miRNA could break through the endosomal barrier, we used the same method as described above to ensure that the p-MSN/miRNA complex was phagocytosed by NRVMs. After 24 hours of incubation, the endolysosomes were labeled with lysotracker Green staining. The miRNA and endolysosome distribution in the cytoplasm was then observed by Zeiss LSM 800 confocal laser microscope(Zeiss zen software ver.2.3) .

## Gel electrophoresis assay

Gel electrophoresis assays were performed using 1×Tris base-acetic acid-ethylenediaminetetraacetic acid (EDTA) buffer containing a 1% (w/v) agarose gel and 1× Goldview nuclear stain (G8142; Solarbio). The miRNA (1 μg) and p-MSN complexes were added to the gel wells at different concentrations and subjected to various treatments. The samples were then pipetted into the gel wells. The electrophoresis was carried out at 100 V for 10 minutes. Finally, the gel was examined using an image analyzer.

## Whole-animal imaging

To monitor the in vivo distribution of miRNAs, we prepared p-MSN/miRNA-cy5(100 μg miRNA in 500 μl mixture per rat systemically) and MSN/miRNA-cy5 complexes and injected them into rats via the tail vein. The rats were then immobilized, and both fluorescence and white light images of the rats were captured using a small animal optical imaging system (MS FX PRO Bruker, Billerica, MA). The intensity of the CY5 signal was measured using a fluorescence camera with an excitation wavelength of 630 nm and an emission wavelength of 700 nm. Additionally, white light images were acquired to aid in identifying the location of the heart. All images were obtained and analyzed using Bruker MI SE software (Bruker Biospin Corporation, Woodbridge, CT, USA).

## Establishment of MI model and tail vein injection of nanoparticles

SD rats (male, 7-8 weeks old, weight 220 ± 20 g) were anesthetized with isoflurane (induced with 3%-4% isoflurane, then maintained with 1.5%-2.5% of isoflurane) and given mechanical ventilation. After left lateral open chest and pericardiectomy, left anterior descending branch ligation was performed. To ensure uniformity in the initial injury among all animals and groups, consistent ligation position and depth were confirmed in each rat. This involved targeting the area between the midpoint of the line connecting the pulmonary artery cones and the lower edge of the left atrium, where coronary arteries and part of the myocardium were ligated with surgical suture needles to a depth of about 2 mm. The p-MSN/NC, MSN/miRmiR199a-5p5, and p-MSN/miR199a-5p complexes (100 μg miRNA in 500 μl of mixture per rat) were prepared and injected via tail vein. On day 7 after ligation, cardiac function was evaluated in SD rats using echocardiography to verify that there were no significant differences in fractional shortening (FS) between the rat groups. The sham-operated group underwent only open-chest surgery without LAD.

## Rat myocardial contractility measurement

After 10 minutes of heparin treatment, rats were euthanized by cervical dislocation in the presence of isoflurane gas overdose, and the hearts were rapidly removed, and the ventricular tissue was perfused and immersed in KH buffer. The preserved ventricular tissue was fixed between two vascular clips, with the apical end fixed and the other end connected to the left ventricle and the echanosensory. The contractile force of the ventricular tissue was recorded by the echanosensory module of the signal acquisition system under 1 Hz electrical stimulation.

## Statistical analysis

Data analysis was conducted using Graphpad prim 8 and SPSS 23.0 software. The distribution of the data was determined by Shapiro Wilk's or Kolmogorov-Smirnov test for normality. Normally distributed data are expressed as mean ± SD, for statistical comparisons between the two groups, an unpaired two-tailed Student's t-test was used. For comparisons between more than two groups, using one-way analysis of variance (ANOVA), followed by Tukey's multiple-comparison post hoc test. Non-normally distributed data are expressed as median and interquartile spacing, and comparisons between groups were made using the Mann-Whitney U test, multiple independent samples using rank sum test. The p-value was calculated, and statistical significance was defined as $p < 0.05$.

## Reporting summary

Further information on research design is available in the Nature Portfolio Reporting Summary linked to this article.

# Data availability

All data supporting the findings from this study are available within the paper and its supplementary information. Figure 1a, registration numbers are GSE114695, GSE208159, GSE24591, GSE114695. Fig. 1g, registration numbers are GSE209991, GSE221780, GSE217771, GSE6174. Raw RNA-sequencing data can be accessed from NCBI SRA database (SRA accession: PRJNA1097501). Any additional raw data will be

available from the corresponding authors upon reasonable request. Source data are provided with this paper.

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

## Author contributions

Conceptualization: Y.C, H.H, X.Q. Methodology: Y.C, S.L, Y.L, H.Y. Investigation: Y.C, S.L, H.H, X.Q. Visualization: Y.C, S.L, Q.L, J.Z. Statistical: Y.C, S.L, Y.L, H.Y. Funding acquisition: H.H, X.Q. Supervision: H.H, X.Q. Picture drawing and design:Y.C, S.L. Writing—original draft: Y.C, S.L. Writing—review & editing: Y.C, H.H, X.Q.

## Funding

This work was supported by the National Natural Science Foundation of China (Grant Nos. U21A20173: X.Q., 52003113: X.Q., 82102228: X.Q., 32071363: X.Q.), Guangdong Basic and Applied Basic Research Foundation (Grant Nos. 2021A1515010745: X.Q., 2020A1515110356: X.Q.), Science and Technology Planning Project of Guangdong Province (2020B1212060037: X.Q.).

## Competing interests

The authors declare no competing interests.
