## [Peer Review File · Nature Communications]

REVIEWER COMMENTS

Reviewer #1 (Remarks to the Author):

Chen et al seek to establish a cardioprotective role of miR-199a-5p during ischemic injury of the heart. The authors identify that miR-199 appears to be downregulated in the infarcted myocardium, corresponding with upregulation of targets such as AGTR1 and MARK4. Using in vitro studies in neonatal myocytes they convincingly demonstrate that AGTR1 and MARK4 are bonafide targets of miR-199, and that miR-199 can prevent their upregulation and associated oxidative stress and contractility deficits following different insults (AngII stimulation, dox treatment) in vitro. The authors also show that miR-199 blocks the stress-induced increase in detyrosinated tubulin, which has been shown to be driven by increased MARK4 activity and to inhibit contractility in failing cardiomyocytes. Finally, the authors demonstrate that a nanoparticle delivery of miR-199 to infarcted rats appears to limit MI-dependent increases in MARK4, AGTR1, detyrosinated tubulin, and to improve several indicators of cardiac function at certain time points following MI.

These are potentially highly significant findings bolstered by novel mechanistic insights, with potentially important translational implications. However, enthusiasm is limited with the current version of this work, as poor data presentation and questionable experimental design make it challenging to properly assess the rigor of the work and to interpret experiments. Specific suggestions are as follows:

Major:

1. In general, a lack of clarity of experimental design and poorly constructed figures hinders the readability, interpretation, and future reproducibility of the work. Please consider generating schematic descriptions to start figures with information regarding timeline, age/sex of animals/cell type, treatment route, dosage/duration, etc. Is miR-199 always administered at the time of infarct, or ever tested where treatment begins after infarction? This is important to clarify as administration at time of MI would be less directly translationally relevant. Additional specific points on this topic:

a. Unjustified use of broken y-axes for data presentation. For example, Fig 1B, C, D, E should be presented on a continuous y-axis for simpler interpretation.

b. Figure 1E, why is miR-140-5p expression not presented in the non-infarcted (normal) heart?

c. Fig. 2A is too small to be interpretable. This is true of other figure panels throughout the manuscript (Fig 4C, Fig 7C bar graphs, for example). In general, figure construction needs to be improved for clarity. Fonts are often distorted and layouts are jumbled, sometimes making it difficult to link a figure element to its appropriate panel and the legend.

d. Figure 3C, the mutagenesis strategy is unclear. The entire sequence displayed seems to be different than the WT AGTR1 3'UTR, not just the miR seed sequence. How much of the mutant 3'UTR

is retained vs. mutated? What specific residues were mutated, and why was the mutagenesis not restricted to the seed sequence? Was this simply a copy/paste error?

e. Figure 4 is largely unreadable due to small/squished fonts.

f. An experimental design schematic for figure 7 would help the reader better understand when rats were infarcted, when treatment was given (and for what duration), and when animals were sacrificed for analysis. Currently the experimental design is obscure. Also, some analysis of statistical significance should be applied to the fractional shortening data in Figure 7F.

g. Please specify throughout whether “cardiomyocytes” refer to neonatal or adult myocytes, which possess important differences. For studies in neonatal myocytes (which I believe may be all in vitro work based on the methods), please use “NRVM” as a more clear and conventional description instead of CM. Also in methods, please provide a reference for “utilizing well-established methods previously reported by our research group”.

2. This point becomes relevant for figure 5 and the conclusion that miR199 increases CM contraction by microtubule detyrosination, which requires further justification. The authors show convincingly that MARK4 is a bonafide target of miR199, and indeed it has been previously demonstrated convincingly that MARK4 deletion reduces dTyr after ischemic injury, as well as that reducing dTyr is sufficient to increase contraction in failing myocytes. The authors are connecting these dots reasonably; however, all that latter work was done in adult cardiomyocytes, which have a very different cytoskeletal architecture than neonatal cardiomyocytes. As such, it is not a given that reducing detyrosinated microtubules is sufficient to increase contractility in DOX induced NRVMs. To support this conclusion, the authors should silence VASH or pharmacologically inhibit dTyr (EPO-Y) and demonstrate protection from DOX induced contractile changes, and loss of miR-199 sensitivity (identical experimental design in figure 5E, but with VASH KD in place of MARK4 KD). This would allow them to conclude that miR-199 improves contractility post-insult by acting through MARK4-dependent regulation of detyrosination, which would provide important mechanistic insight that should then be articulated more clearly, including in the abstract.

Reviewer #2 (Remarks to the Author):

The authors present a comprehensive set of studies, and use a novel nanoparticle delivery system to highlight the therapeutic potential of targeting miR199-5p in a setting of myocardial infarction. To better evaluate the potential significance of the current results, the authors should address the queries listed below.

1. Lines 117-118. It is noted that miR199-5a and miR-140-5p were widely present in infarcted hearts and either decreased or increased. Of the 4 studies presented in Fig 1A, in which studies was 199a-5p increased vs decreased? And does this depend on the timeframe/remodeling post-MI?

2. After LAD ligation, how was the reproducibility of the procedure verified to ensure the initial injury was consistent between animals and groups prior to an intervention? Some investigators do this by performing an echocardiography assessment and ECG/EKG after the surgery before the intervention.

3. Improving the presentation of some data:

a) For all graphs, scatter plots/individual data points should be presented. This has been done for most graphs but not all. For example: Fig 1F is presented correctly but Fig 1D-E are not. The same applies to the supp section e.g. Fig S6A is presented correctly but Fig S6B-C are not. Please ensure individual data points are provided for all graphs within the manuscript.

b) For some graphs, breaks in the axis should not be used, as the location of the break could lead to misinterpretation of the data. This includes individual data points from within the same group that are spread across the break in axis for example: Fig 1B, Fig 1C, Fig 1E. Further, for these groups, if the data are not normally distributed, non-parametric analyses should be used and the statistics used presented in fig legends.

c) Western blots- molecular weight markers should be presented on all blots (correct for Fig S5G but not other westerns), and uncropped western images should be provided within the data supplement. Scatter plots/individual data points within the bar graphs should be presented. This applies to multiple figures e.g. Fig 3, Fig 4, Fig 5, Fig 7.

4. For all in vivo experiments with assessment of cardiac function by echocardiography data, a supplementary table/fig should be provided showing heart rates of each group at each time point, as well as LV volume measures used to derive FS and EF. Statistics should be added to Fig 7E. I assume this is paired analysis. Given FS is highly dependent on HR, the same graph for HR should be presented in the Supplement.

5. For morphological data referencing heart weight/body weight, the body weights of each group before and after MI should also be presented in a supplementary table/fig. MI models often lose body weight as LV remodeling and heart failure progresses. A better measure is HW/tibia length. The same comment applies to lung weight/body weight in assessing lung congestion.

6. How much was miR199-5a increased in the infarct area of the miR199-5a mimic group vs the infarct area of the NC group at 6 months and 12 months for the animals presented in Fig 2? Were the validated miR199-5a targets (AGTR1, MARK4) also regulated in the infarcted tissue in the animals in Fig 2?

7. P-MSN/miR199a-5p – in vivo 4 week data after MI (Fig 7): did this approach have an impact on heart and lung weights as shown with the direct injection in Fig 2?

8. It appears the study was only performed on male rats. Why were female rats not included in this study?

9. Methods- Further details are required such as:

a) percentage of isoflurane used for echocardiography and the MI surgery

b) agent, route of administration etc for euthanasia for each study.

c) details of antibodies including antibody ID numbers, concentrations etc

9. Statistical analysis- It appears that some data are not normally distributed. In these instances, non-parametric tests should be performed.

Minor comments

-Fig 1C- For clarity insert * vs Normal. Fig 1F- Figure legend describes groups as adult cardiomyocytes and non-cardiomyocytes. Define CF- cardiac fibroblasts.

-Fig 2: Panel A: Graph is too small and blurry to see, background color for box "pathology analysis" is quite dark, hard to see the text. Overall quality of the experimental time line is not reproducing well.

-Fig 4: -Labels on many of the figures are very small and illegible on a print out e.g. labels on some graphs in Fig 4 and statistics symbols. Please enlarge.

-Fig 6: Ideally add MWs to panel B or insert them into the legend.

- Figure 7: X-axis for most graphs difficult to see (Fig 7C in particular) or appear stretched out. Fig 7C images should have a scale bar inserted.

-The construction strategy of P-MSNs/miR199-5p (top panel of current Fig 8) could be moved to Fig 6 as panel A or moved to the supplement.

-line 343: reference to collagen markers. Fig S6C rather than Fig S4C should be referenced.

-Ideally western blots and quantitation would be presented together rather than the western image in the main paper and quantitation in the Supp material e.g. Fig 4F and Fig S3 C

Reviewer #3 (Remarks to the Author):

In this manuscript, the authors discovered that downregulation of miR199a-5p in ischemic heart contributed to myocardial infarction through concerted mechanisms, and developed a delivery system achieving single-dose infusion for long-term therapeutic efficacy. The findings are novel and of translational potential. However, the design of the delivery system is insufficient, and the efficacy and selectivity cannot be guaranteed. The below concerns are raised.

Major concerns:

1. There are no investigations on the targeting efficiency or specificity of the delivery system. Although some data on biodistribution are provided, mechanisms for heart engraftment are lacking. This reviewer even thinks that there is no heart tropism for the delivery system.

2. There are no explanations on why the delivery system is predominantly localized to damaged cardiomyocytes rather than damaged non-cardiomyocytes, and damaged cardiomyocytes rather than healthy cardiomyocytes.

3. There are no clues on how the nanoparticles may really escape from endosomes and lysosomes. Also no evidence is provided on the microRNA integrity after the claimed escape inside the recipient cells.

4. What is the half-life of miR199a-5p? For a single-dose but long-term therapeutic, either the delivered molecule or the regulated cascade has the potential of lasting effects. This reviewer also does not think the revealed mechanistic targets have lasting regulatory potential. No epigenetic or metabolic research has been performed.

Minor concerns:

1. Fluorescence in situ hybridization should be performed on heart tissues for confirming the microRNA expression.

2. Histological examinations should be performed to better show the details of infarcted tissues. IF staining of certain protein markers should also be added.

3. Would miR199a-5p be released by cells in forms of extracellular vesicles? Please discuss any potential side-effects or factors affecting the robustness of the methodology.

Detailed responses to the reviewers' comments

We appreciate the reviewers' insightful comments and value the expert recommendations provided. To enhance the robustness of our manuscript, we have conducted supplementary experiments and incorporated novel data in the revised version. In this response, we systematically address the reviewers' concerns and recommendations, point by point.

Reviewer #1 (Remarks to the Author):

Chen et al seek to establish a cardioprotective role of miR-199a-5p during ischemic injury of the heart. The authors identify that miR-199 appears to be downregulated in the infarcted myocardium, corresponding with upregulation of targets such as AGTR1 and MARK4. Using in vitro studies in neonatal myocytes they convincingly demonstrate that AGTR1 and MARK4 are bonafide targets of miR-199, and that miR-199 can prevent their upregulation and associated oxidative stress and contractility deficits following different insults (AngII stimulation, dox treatment) in vitro. The authors also show that miR-199 blocks the stress-induced increase in detyrosinated tubulin, which has been shown to be driven by increased MARK4 activity and to inhibit contractility in failing cardiomyocytes. Finally, the authors demonstrate that a nanoparticle delivery of miR-199 to infarcted rats appears to limit MI-dependent increases in MARK4, AGTR1, detyrosinated tubulin, and to improve several indicators of cardiac function at certain time points following MI.

These are potentially highly significant findings bolstered by novel mechanistic insights, with potentially important translational implications. However, enthusiasm is limited with the current version of this work, as poor data presentation and questionable experimental design make it challenging to properly assess the rigor of the work and to interpret experiments. Specific suggestions are as follows:

Responses: We appreciate the reviewers for their positive and insightful evaluation of

our work. In the revised manuscript, schematic diagrams have been incorporated into Figures 2, 6, and 7, detailing aspects such as the timeline, animal age/sex, treatment route, dose, and nanoparticle synthesis method. Molecular weight markers have been included in the gel plots, statistical graphs have been adjusted to ensure continuous axes, individual data points have been plotted on the bar graphs, and the layout and font size of Figures 2, 4, and 7 have been modified to enhance readability. As to your mentioned shortcomings in the initial version in terms of data presentation and experimental design, we have addressed all of these issues in the revisions and additional content, making our article more comprehensible, analyzable, and replicable for future studies.

Major:

1. In general, a lack of clarity of experimental design and poorly constructed figures hinders the readability, interpretation, and future reproducibility of the work. Please consider generating schematic descriptions to start figures with information regarding timeline, age/sex of animals/cell type, treatment route, dosage/duration, etc.

Responses: We thank the reviewer for this advice. In the revised manuscript, we included schematic diagrams in Figures 2, 6, and 7 to present details such as the timeline, animal age/sex, treatment route, dosage, and nanoparticle synthesis method. Additionally, descriptions of cell types were incorporated into Figures 3, 4, and 5.

Is miR-199 always administered at the time of infarct, or ever tested where treatment begins after infarction? This is important to clarify as administration at time of MI would be less directly translationally relevant.

Responses: Thanks for your advice. In Figure 2, miR-199 was delivered via intramyocardial injection following left anterior descending ligation, while in Figure 7, miR-199 was administered via caudal vein injection after completing left anterior descending ligation and suturing the chest cavity. Actually, miR-199 was administered immediately after infarct modeling in all cases, assess of the miR-199 levels in the infarcted heart post-modeling and before injection is difficult. However, we

demonstrated a significant decrease in miR-199 within 1 week after infarction by analyzing the GEO database and assaying miR-199 levels in the infarcted area of rats (Fig. 1a, d).

To test how long miR199a was enriched in cells after injection of miR199a nanoparticles, we tail vein injected miR-199a-cy5 nanoparticles to observe the fluorescence changes in the precordial region, and the results showed that the cy5 fluorescence signal in the precordial region was enriched after 2 hours of administration of P-MSN/miR-199a-5p-cy5 nanoparticles (Fig. 6b). RT-qPCR analysis revealed a significant upregulation of miR-199a-5p expression in the infarcted area. After 3 days, miR-199a-5p expression showed a nearly 5-fold increase compared to the distal region, which further escalated to approximately 15-fold at day 7 and maintained a 5-fold increase by week 4 (Fig S5a). *In vitro*, 1 hour post-treatment with P-MSN/miR-199a-5p nanoparticles in primary cardiomyocytes, miR-199a-5p levels was surged by around 15-fold and sustained this elevation for 12 hours (Fig S4e). These findings suggest that P-MSN/miR-199a-5p nanoparticles exhibit rapid enrichment in the precordial region 2 hours post-injection, along with efficient cellular uptake within 1 hour and sustained presence for up to 4 weeks.

Additional specific points on this topic:

a. Unjustified use of broken y-axes for data presentation. For example, Fig 1B, C, D, E should be presented on a continuous y-axis for simpler interpretation.

Responses: Thank you for your feedback. Following your suggestions, we have revised the labeling of the y-axis. Figures 1b, 1c, 1d, and 1e now feature a continuous y-axis representation.

b. Figure 1E, why is miR-140-5p expression not presented in the non-infarcted (normal) heart?

Responses: In response to the reviewer's suggestion, we elevated miR-140-5p levels in non-infarcted (normal) hearts in Figure 1e.

Fig1e

c. Fig. 2A is too small to be interpretable. This is true of other figure panels throughout the manuscript (Fig 4C, Fig 7C bar graphs, for example). In general, figure construction needs to be improved for clarity. Fonts are often distorted and layouts are jumbled, sometimes making it difficult to link a figure element to its appropriate panel and the legend.

Responses: We sincerely apologize for the error and any inconvenience it may have caused. In the revised version, we have reuploaded Figures 2, 4, and 7, and enhanced the fonts and images to improve their interpretability.

d. Figure 3C, the mutagenesis strategy is unclear. The entire sequence displayed seems to be different than the WT AGTR1 3'UTR, not just the miR seed sequence. How much of the mutant 3'UTR is retained vs. mutated? What specific residues were mutated, and why was the mutagenesis not restricted to the seed sequence? Was this simply a copy/paste error?

Responses: We sincerely apologize for this copy/paste error and express our gratitude to the reviewers for their meticulous reading in identifying it. The adjusted mutated AGTR1 3'UTR sequence has been integrated into Figure 3c.

e. Figure 4 is largely unreadable due to small/squished fonts.

Responses: We deeply apologize for the mistake and the inconvenience caused. We have reuploaded Figures 4 in the revised version and enlarged the fonts and images to make them more readable.

f. An experimental design schematic for figure 7 would help the reader better understand when rats were infarcted, when treatment was given (and for what duration), and when animals were sacrificed for analysis. Currently the experimental design is obscure. Also, some analysis of statistical significance should be applied to the fractional shortening data in Figure 7F.

Responses: We thank the reviewer for this suggestion. As advised, we have augmented the schematic of the experimental design in Figure 7 to include details such as the sex and weight of the rats, the timing of infarction, treatment administration (therapeutic dose), and the time of animal sacrifice for testing (Fig. 7a). Furthermore, we have conducted statistical analysis on the fractional shortening data presented in Figure 7f for significance.

g. Please specify throughout whether “cardiomyocytes” refer to neonatal or adult myocytes, which possess important differences. For studies in neonatal myocytes (which I believe may be all in vitro work based on the methods), please use “NRVM” as a more clear and conventional description instead of CM. Also in methods, please provide a reference for “utilizing well-established methods previously reported by our research group”.

Responses: Thanks for your advice. Following your suggestion, we utilized “NRVM” instead of “CM.” In the methodology section, we referenced three publications from our research group that detail the extraction of primary cardiomyocytes. The citations are as follows:

1. WANG L, LIU Y, YE G, et al. Injectable and conductive cardiac patches repair infarcted myocardium in rats and minipigs [J]. Nat Biomed Eng, 2021, 5(10):

1157-73.

2. SONG C, KONG F, NONG H, et al. Ammonium Persulfate-Loaded Carboxylic Gelatin-Methacrylate Nanoparticles Promote Cardiac Repair by Activating Epicardial Epithelial-Mesenchymal Transition via Autophagy and the mTOR Pathway [J]. ACS Nano, 2023, 17(20): 20246-61.

3. HE Y, LI Q, CHEN P, et al. A smart adhesive Janus hydrogel for non-invasive cardiac repair and tissue adhesion prevention [J]. Nat Commun, 2022, 13(1): 7666.

2. This point becomes relevant for figure 5 and the conclusion that miR199 increases CM contraction by microtubule detyrosination, which requires further justification. The authors show convincingly that MARK4 is a bonafide target of miR199, and indeed it has been previously demonstrated convincingly that MARK4 deletion reduces dTyr after ischemic injury, as well as that reducing dTyr is sufficient to increase contraction in failing myocytes. The authors are connecting these dots reasonably; however, all that latter work was done in adult cardiomyocytes, which have a very different cytoskeletal architecture than neonatal cardiomyocytes. As such, it is not a given that reducing detyrosinated microtubules is sufficient to increase contractility in DOX induced NRVMs. To support this conclusion, the authors should silence VASH or pharmacologically inhibit dTyr (EPO-Y) and demonstrate protection from DOX induced contractile changes, and loss of miR-199 sensitivity (identical experimental design in figure 5E, but with VASH KD in place of MARK4 KD). This would allow them to conclude that miR-199 improves contractility post-insult by acting through MARK4-dependent regulation of detyrosination, which would provide important mechanistic insight that should then be articulated more clearly, including in the abstract.

Responses: We highly appreciate the valuable suggestion provided by the reviewer. MARK4 plays a key role in facilitating the entry of VASH2 into microtubules to regulate myocardial contractility. In this study, we designed three siRNAs to target the VASH2 gene in rat NRVMs. Recognizing the potential impact of the VASH2 gene on cell viability, we performed CCK-8 assays to evaluate the viability of NRVMs in the

transfected control and various siVASH2 groups. The results indicated no significant difference in viability among the groups (Fig.S3b). Subsequently, Western blot analysis was conducted to assess the silencing efficiency of the three siRNAs. The data revealed a considerable reduction in VASH2 protein expression with the introduction of the 2# siVASH2 in NRVMs (Fig.S3c). Therefore, we chose the 2# siVASH2 for the subsequent cardiomyocyte contractility analysis. This assessment demonstrated that DOX treatment led to a decline in heartbeat rate and amplitude in NRVMs compared to the control baseline. Treatment with the miR199a-5p mimic preserved the decrease in heartbeat rate and amplitude, while inhibition of miR199a-5p exacerbated these reductions. Notably, following VASH2 silencing, the regulatory effect of miR199a-5p on NRVM contractility disappeared (Fig.S3d).

Reviewer #2 (Remarks to the Author)

The authors present a comprehensive set of studies, and use a novel nanoparticle delivery system to highlight the therapeutic potential of targeting miR199-5p in a setting of myocardial infarction. To better evaluate the potential significance of the current results, the authors should address the queries listed below.

Responses: We would like to thank the Reviewer for this positive and thoughtful summation of our work. We have conducted supplementary experiments according to your kind suggestions.

1. Lines 117-118. It is noted that miR199-5a and miR-140-5p were widely present in infarcted hearts and either decreased or increased. Of the 4 studies presented in Fig 1A, in which studies was 199a-5p increased vs decreased? And does this depend on the timeframe/remodeling post-MI?

Responses: In the GSE24591 study, miR199a-5p decreased at 6 hours after infarction, while in the GSE114695 study, it decreased 1 week after the event. Conversely, miR199a-5p increased 8 weeks post-infarction in the same studies, as observed in the GSE208159 study where it increased 4 weeks post-infarction. To further investigate the fluctuating pattern of miR199a-5p following infarction, we examined additional miR199a-5p alterations at different time points post-infarction within the GEO database. Our analysis revealed a decrease in miR199a-5p levels after 1 day (GSE138141) and 5 days (GSE53211) post-infarction, with similar reductions observed at 4 weeks (GSE124545) and 2 months (GSE18129) post-infarction. Remarkably, in patients who suffered sudden death due to infarction, miR199a-5p showed a significant decline in the infarcted area compared to the non-infarcted region. Based on these observations, we hypothesized that miR199a-5p experiences a substantial decrease after 1 week following infarction, potentially linking it to sudden cardiac death. Additionally, we suggested individual variability in miR199a-5p changes between 4 to 8 weeks post-infarction.

2. After LAD ligation, how was the reproducibility of the procedure verified to ensure the initial injury was consistent between animals and groups prior to an intervention? Some investigators do this by performing an echocardiography assessment and ECG/EKG after the surgery before the intervention.

Responses: To ensure uniformity in the initial injury among all animals and groups, male rats weighing approximately 220 g were selected, and consistent ligation

position and depth were confirmed in each rat. This involved targeting the area between the midpoint of the line connecting the pulmonary artery cones and the lower edge of the left atrium, where coronary arteries and part of the myocardium were ligated with surgical suture needles to a depth of about 2 mm. Echocardiography was conducted on day 7 post-myocardial infarction to verify that there were no significant differences in fractional shortening (FS) between the rat groups (Fig.7f).

3. Improving the presentation of some data:

a) For all graphs, scatter plots/individual data points should be presented. This has been done for most graphs but not all. For example: Fig 1F is presented correctly but Fig 1D-E are not. The same applies to the supp section e.g. Fig S6A is presented correctly but Fig S6B-C are not. Please ensure individual data points are provided for all graphs within the manuscript.

Responses: Thanks for your advice. We have made the necessary changes to replace the diagrams in the manuscript with scatter plots or individual data points for the following figures: Figures 1d, 1e; Figure 3d, 3h; Figures 4c, 4e, 4f, 4h, 4i; Figures 5a, 5c, 5e; Figure 6d; Figures 7a, 7b, 7c, 7F; Figures S2a, S2b; Figures S3a, S3b; Figures S4a, S4c; Figures S5b, S5c. These modifications will provide a clearer visualization of the data presented in the manuscript.

b) For some graphs, breaks in the axis should not be used, as the location of the break could lead to misinterpretation of the data. This includes individual data points from within the same group that are spread across the break in axis for example: Fig 1B, Fig 1C, Fig 1E.

Responses: We regret any confusion that may have arisen. The y-axis labeling for the specified figures in the manuscript has been revised. Figures 1b, 1c, 1d, and 1e now utilize a continuous y-axis, in accordance with your guidelines.

c) Western blots- molecular weight markers should be presented on all blots (correct for Fig S5G but not other westerns), and uncropped western images should be

provided within the data supplement. Scatter plots/individual data points within the bar graphs should be presented. This applies to multiple figures e.g. Fig 3, Fig 4, Fig 5, Fig 7.

Responses: Following the reviewer's recommendation, molecular weight markers were included on the Western blots of Figure 3d, h, Figure 4f, i, Figure 5b-d, and Figure 7c. Additionally, uncropped Western blot images were provided in the raw data. Furthermore, various charts in the manuscript were replaced to display scatter plots or individual data points, specifically in Figs. 1d-e, Fig. 3d, h, Fig. 4c, e, f, h, i, Fig. 5a, c, e, Fig. 6d, Fig. 7b-d, Fig. S2a-e, Fig. S3a-b, Fig. S4a, c, Fig. S5b-c.

4. For all in vivo experiments with assessment of cardiac function by echocardiography data, a supplementary table/fig should be provided showing heart rates of each group at each time point, as well as LV volume measures used to derive FS and EF. Statistics should be added to Fig 7E. I assume this is paired analysis. Given FS is highly dependent on HR, the same graph for HR should be presented in the Supplement.

Responses: We thank the reviewers for this suggestion. Following their advice, we have incorporated echocardiographic data including rat heart cardiac function indices: heart rate (HR), left ventricular ejection fraction (LVEF), left ventricular short-axis shortening (LVFS), diastolic and systolic volumes (LV Vol;d, LVVVol;s), and left ventricular mass (LV Mass); and cardiac structural indices: LV posterior wall thickness at end-diastole and end-systole (LVPW;d, LVPW;s), diastolic LV anterior wall thickness at end-diastole and end-systole (LVAW;d, LVAW;s), LV internal diameter at end-diastole and end-systole (LVID;d, LVID;s) for both the 1-year (Table 1) and 4-week (Table 2) animal models. Furthermore, we conducted statistical analysis on the fractional shortening data presented in Figure 7e to determine its significance, as illustrated in Figure 7f.

5. For morphological data referencing heart weight/body weight, the body weights of each group before and after MI should also be presented in a supplementary table/fig.

MI models often lose body weight as LV remodeling and heart failure progresses. A better measure is HW/tibia length. The same comment applies to lung weight/body weight in assessing lung congestion.

Responses: In line with the reviewer's recommendation, heart weight/body weight was replaced with heart weight/tibia length in Figures 2d and 2h, and the body weights, heart weights, and lung weights of rats post-myocardial infarction were included in the raw data. As for the body weight of pre-myocardial infarction rats, to ensure consistency, we maintained a body weight range of $220 \text{ g} \pm 20 \text{ g}$ for the pre-myocardial infarction modeling rats.

6. How much was miR199-5a increased in the infarct area of the miR199-5a mimic group vs the infarct area of the NC group at 6 months and 12 months for the animals presented in Fig 2? Were the validated miR199-5a targets (AGTR1, MARK4) also regulated in the infarcted tissue in the animals in Fig 2?

Responses: We appreciate the valuable suggestion from the reviewers. Following this recommendation, we assessed the levels of miR199a-5p at 6 and 12 months using the FISH technique. The findings revealed an elevation in miR199a-5p levels in the rat infarct region within the miR199a-5p group compared to the negative control group at both time points (Fig.S1c). Additionally, we employed immunofluorescence to detect the expression of AGTR1 and MARK4 in the infarct region of rats at 6 and 12 months. The results demonstrated a reduction in the expression of AGTR1 and MARK4 in the miR199a-5p group in comparison to the negative control group (Fig.S1c).

FigS1c

7. P-MSN/miR199a-5p – in vivo 4 week data after MI (Fig 7): did this approach have an impact on heart and lung weights as shown with the direct injection in Fig 2?

Responses: In response to the reviewer’s suggestion, we evaluated the impact of P-MSN/miR199a-5p nanoparticles on the heart and lung weights following a 4-week treatment of rats with myocardial infarction. The results indicated that rats injected with P-MSN/miR199a-5p and MSN/miR199a-5p exhibited decreased heart weight/body weight ratios, heart weight/tibia length ratios, and lung weight/body weight ratios compared to the control group. Furthermore, the reductions were more pronounced with P-MSN/miR199a-5p than with MSN/miR199a-5p (Fig.S5c).

FigS5c

8. It appears the study was only performed on male rats. Why were female rats not

included in this study?

Responses: To ensure consistency in the role of miRNA without potential sex-related variations, it would have been ideal to utilize an equal distribution of male and female rats for myocardial infarction modeling. However, due to the prolonged duration of the research where the rats were observed for one year post-myocardial infarction, starting at approximately 2-3 months old and reaching around 14-15 months old, certain considerations arose. Female rats typically enter perimenopause around 12 months and experience menopause between 15 to 18 months. Understanding the potential impact of menopause on ventricular and vascular function^[1], we opted to use male rats as the model subjects for this study to maintain consistency and limit the confounding effects of hormonal changes related to menopause on the outcomes.

[1] HAYWARD C S, KELLY R P, COLLINS P. The roles of gender, the menopause and hormone replacement on cardiovascular function [J]. *Cardiovasc Res*, 2000, 46(1): 28-49.

9. Methods- Further details are required such as:

- a) percentage of isoflurane used for echocardiography and the MI surgery
- b) agent, route of administration etc for euthanasia for each study.
- c) details of antibodies including antibody ID numbers, concentrations etc

Responses: Thanks to the reviewers' advice, the revised version now includes details regarding the dosage of anesthetics, euthanasia procedures, and the ID numbers and concentration of antibodies, as recommended by the reviewers. The details are listed as following:

- a) Rats undergoing myocardial infarction (MI) surgery were initially induced with 3%-4% isoflurane and subsequently maintained on 1.5%-2.5% isoflurane. For rats undergoing echocardiography, the induction involved 3% isoflurane, followed by maintenance on 2% isoflurane.
- b) In the event of isoflurane gas overdose, the rats were humanely euthanized by cervical dislocation, following which the hearts were promptly excised for further analysis.
- c) The following antibodies were used in the study with their respective stock

numbers, applications, and recommended concentrations:

- Angiotensin II Type 1 Receptor (AGTR1, Abcam, ab239995, WB recommended concentration: 1 µg/ml)
- Cardiac troponin T (cTNT, Abcam, ab10214, IF recommended concentration: 1 µg/ml)
- α -actinin (Abcam, ab9465, IF recommended concentration: 2 µg/ml)
- 8-OHG (Abcam, ab62623, IF recommended concentration: 2 µg/ml)
- Alpha Tubulin (α -tub, Abcam, ab7291, WB recommended concentration: 1 µg/ml)
- NADPH oxidase 4 (NOX4, Abcam, ab154244, WB recommended concentration: 1 µg/ml)
- Detyrosinated alpha Tubulin (dTyr-tub, Abcam, ab254154, WB recommended concentration: 1 µg/ml)
- MARK4(Cell Signaling Technology,4834,WB recommended concentration: 1 µg/ml)

10. Statistical analysis- It appears that some data are not normally distributed. In these instances, non-parametric tests should be performed.

Responses: In response to the reviewers' feedback, the data in Figs. 1B, C, and D were determined to exhibit non-normal distribution and were consequently analyzed using nonparametric statistical methods, as indicated in the legend and methods sections. Variables with skewed distributions were presented as median and interquartile range. The Mann-Whitney U test was employed for comparing two groups, while the rank sum test was utilized for comparisons involving multiple independent samples.

Minor comments

-Fig 1C- For clarity insert * vs Normal. Fig 1F- Figure legend describes groups as adult cardiomyocytes and non-cardiomyocytes. Define CF- cardiac fibroblasts.

Responses: Incorporating the reviewers' suggestions, we have included delineating

lines in Figure 1C to clarify the groups earmarked for statistical analysis. Additionally, the figure legend in Figure 1F has been revised from “noncardiac myocytes” to “cardiac fibroblasts” as advised.

-Fig 2: Panel A: Graph is too small and blurry to see, background color for box “pathology analysis” is quite dark, hard to see the text. Overall quality of the experimental time line is not reproducing well.

Responses: We deeply apologize for the mistake and the inconvenience caused. We have reuploaded all Figures in the revised version and enlarged the fonts and images to make them more readable.

-Fig 4: -Labels on many of the figures are very small and illegible on a print out e.g. labels on some graphs in Fig 4 and statistics symbols. Please enlarge.

Responses: We regret the errors and any inconvenience caused. In this revised version, we have increased the font size and adjusted the image sizes to enhance readability.

-Fig 6: Ideally add MWs to panel B or insert them into the legend.

Responses: Following the reviewer’s suggestion, we have added molecular weights in Figure 6b for improved clarity and accuracy.

- Figure 7: X-axis for most graphs difficult to see (Fig 7C in particular) or appear stretched out. Fig 7C images should have a scale bar inserted.

Responses: We are sorry for the errors made and the inconvenience caused. In the updated version, we have increased the font size of the axes for most of the images, including Figure 7, and included a scale bar in Figure 7C for better clarity and precision.

-The construction strategy of P-MSNs/miR199-5p (top panel of current Fig 8) could be moved to Fig 6 as panel A or moved to the supplement.

Responses: In accordance with the reviewer's suggestion, we have relocated the schematic of the P-MSNs/miR199-5p construction in Fig. 8, now designated as Fig. 6a.

-line 343: reference to collagen markers. Fig S6C rather than Fig S4C should be referenced.

Responses: We apologize profusely for this error and thank the reviewers for their careful reading to discover it. In the updated version, we have corrected the reference to collagen markers in line 343 to refer to Fig S6C.

-Ideally western blots and quantitation would be presented together rather than the western image in the main paper and quantitation in the Supp material e.g. Fig 4F and Fig S3 C.

Responses: Following the reviewer's suggestion, all Western blot images and their quantifications were presented in a single figure for clearer visualization.

Reviewer #3 (Remarks to the Author):

In this manuscript, the authors discovered that downregulation of miR199a-5p in ischemic heart contributed to myocardial infarction through concerted mechanisms, and developed a delivery system achieving single-dose infusion for long-term therapeutic efficacy. The findings are novel and of translational potential. However, the design of the delivery system is insufficient, and the efficacy and selectivity cannot be guaranteed. The below concerns are raised.

Responses: We thank the reviewers for acknowledging our work. In our research, we screened for regular changes in miR199a-5p by analyzing and validating the altered miRNA after myocardial infarction. miR199a-5p significantly decreased one week after infarction, and the low levels of miR199a-5p within a week may be related to sudden cardiac death. Conversely, at 4 and 8 weeks post-infarction, miR-199a-5p levels exhibited a gradual increase, accompanied by noticeable individual variations.

Mechanistic studies have shown that miR199a-5p plays a protective role in myocardial infarction by targeting different biological processes at various stages. In the early stages, miR199a-5p reduces cardiomyocyte apoptosis and improves heart function by targeting AGTR1. In the later stages, miR199a-5p enhances myocardial contractility and mitigates post-infarction heart failure by targeting MARK4. The dual protective mechanism of miR-199a-5p offers promise for long-term repair strategies in myocardial infarction.

In order to exert the dual protective effects of miR-199a-5p in the long term, an efficient delivery system with nucleic acid protection, high drug-carrying capacity, and long-term stability is essential, and the system should be tailored to the changes in miR-199a-5p levels patterns after infarction. Considering the limitations in sustained gene expression using viral vectors (PMID: 24855205) and the advantageous attributes of MSNs such as enhanced drug-carrying capacity and stability compared to “softer” materials like liposomes or polymers (PMID: 37774811), MSNs with specific dimensions and properties were chosen. These MSNs, characterized by a diameter of 160 nm, a specific surface area of 410-680 m²/g, a pore size of 2.8-13.3 nm, and a pore volume of 0.57-1.66 cm³/g, demonstrated a drug loading capacity of 12.3%. The incorporation of polyethyleneimine (PEI) increased this capacity to 16.6%, enhancing miRNA encapsulation efficiency and promoting lysosomal escape through proton sponging (Fig. 6c). The pore size structure of 2.8-13.3 nm in MSN facilitated controlled release of microRNA therapeutics, with 80% of the drug being released after 3 days (Fig. S4b) while protecting microRNA from nuclease degradation (Fig. 6b).

Regarding the selectivity of the ischemic heart, our delivery system primarily utilizes the PEP peptide, a cyclic sequence of nine amino acids (CSTSMKAC), discovered by Kanki S et al. in 2011 (PMID: 21316369). Multiple studies have confirmed that this peptide preferentially acts on ischemic myocardium rather than non-ischemic left ventricular and non-coronary artery cells (PMID: 27107168, PMID: 30371236, PMID: 36625783). Additionally, commercial PEP peptides targeting ischemic myocardial cells can be obtained through MedChemExpress (Cat No:

HY-P5217).

Both *in vivo* and *in vivo* experiments demonstrated that our constructed P-MSN/miR199a-5p nanoparticles promoted a significant increase in miR199a-5p levels within 1 week, which lasted up to 4 weeks (Fig.S4i, S5a). P-MSN/miR199a-5p nanoparticles can effectively match the changing pattern of miR199a-5p post-infarction, thus exerting better myocardial infarction repair effects.

Major concerns:

1. There is no investigations on the targeting efficiency or specificity of the delivery system. Although some data on biodistribution are provided, mechanisms for heart engraftment are lacking. This reviewer even thinks that there is no heart tropism for the delivery system.

Responses: We are thankful to the reviewers for their comments. As direct fluorescence detection after miRNA labeling with fluorescent markers may not accurately reflect changes in miR199a-5p expression in cells and tissues, we chose to assess the transfection efficiency of nanoparticles using RT-qPCR in both *in vivo* and *in vitro* settings. In the *in vitro* experiments, primary cardiomyocytes were treated with or without H₂O₂ and incubated with P-MSN/NC, MSN/miR199a-5p, or P-MSN/miR199a-5p for 12 hours. The RT-qPCR analysis revealed no significant difference in the intracellular miR199a-5p expression between the MSN/miR199a-5p and P-MSN/miR199a-5p groups under normal conditions. However, following cell injury induced by H₂O₂, miR199a-5p expression was enhanced 1-fold in the P-MSN/miR199a-5p group compared to the MSN/miR199a-5p group (Fig.S4k).

In the *in vivo* study, we investigated miR199a-5p expression in the distal and infarcted regions of rats after injecting P-MSN/miR199a-5p into the tail vein of infarcted rats for 3 days, 7 days, and 4 weeks. The RT-qPCR results demonstrated a notable increase in miR199a-5p expression in the infarcted region, with approximately a 5-fold rise after 3 days compared to the distal region. This elevation further increased to around 15-fold at 7 days and was maintained at a 5-fold increase by the 4th week (Fig.S5a).

FigS4k

FigS5a

2. There is no explanations on why the delivery system is predominantly localized to damaged cardiomyocytes rather than damaged non-cardiomyocytes, and damaged cardiomyocytes rather than healthy cardiomyocytes.

Responses: The delivery system we constructed mainly targets damaged cardiomyocytes through PEP peptide, a 9-amino-acid cyclic sequence (CSTSMKAC), which was discovered in ischemic myocardium by Kanki S, et al. through phage experiments in 2011^[1] and has been demonstrated in several studies to preferentially target cardiomyocytes in ischemic injury myocardium^[2-4], and MedChemExpress (Cat No: HY-P5217) also has commercialized PEP peptide for targeting cardiomyocytes in ischemically injured myocardium. Despite its effectiveness, the precise mechanism through which the PEP peptide binds to ischemic myocardium remains unknown. It is theorized that the peptide may mimic natural ligands that bind to receptors exposed to ischemic injury, such as 34834-STSKMLKA-34841 (titin), 531-TSMLKA-536 (optic atrophy-1, OPA-1), or 716-MLKA-719 (dynamin-1 like protein, DRP-1).

The cytoskeletal protein titin has been identified as a protein with sequence similarity to the peptide CSTSMKAC. In ischemic conditions, the binding of alpha B-crystallin in cardiomyocytes to the I-band portion of cardiac titin is promoted,

which helps stabilize titin and protect it from degeneration due to ischemic injury^[5]. This observation leads to the hypothesis that the concentration of CSTSMLKAC in ischemic cardiomyocytes may be facilitated through its direct interaction with alpha B-crystallin. Extensive research is necessary to elucidate the precise mechanism underlying the binding of the PEP peptide to ischemic myocardium.

- [1] KANKI S, JAALOUK D E, LEE S, et al. Identification of targeting peptides for ischemic myocardium by in vivo phage display [J]. *J Mol Cell Cardiol*, 2011, 50(5): 841-8.
- [2] FERREIRA M P, RANJAN S, CORREIA A M, et al. In vitro and in vivo assessment of heart-homing porous silicon nanoparticles [J]. *Biomaterials*, 2016, 94: 93-104.
- [3] WANG X, CHEN Y, ZHAO Z, et al. Engineered Exosomes With Ischemic Myocardium-Targeting Peptide for Targeted Therapy in Myocardial Infarction [J]. *J Am Heart Assoc*, 2018, 7(15): e008737.
- [4] SUN X, CHEN H, GAO R, et al. Intravenous Transplantation of an Ischemic-specific Peptide-TPP-mitochondrial Compound Alleviates Myocardial Ischemic Reperfusion Injury [J]. *ACS Nano*, 2023, 17(2): 896-909.
- [5] GOLENHOFEN N, ARBEITER A, KOOB R, et al. Ischemia-induced association of the stress protein alpha B-crystallin with I-band portion of cardiac titin [J]. *J Mol Cell Cardiol*, 2002, 34(3): 309-19.

3. There is no clues on how the nanoparticles may really escape from endosomes and lysosomes. Also no evidence is provided on the microRNA integrity after the claimed escape inside the recipient cells.

Responses: Thanks for your advice. The lysosomal escape phenomenon arises mainly due to changes in osmotic pressure and membrane solubilization activity^[1]. Our delivery system induces lysosomal escape through two mechanisms involving PEI. First, PEI carries a strong positive charge, which facilitates cellular internalization by using the positive surface charge to interact with negatively charged lysosomal membranes to reduce the stability of lysosomal membranes^[2,3]. In addition, the backbone of PEI contains one nitrogen atom for every three atoms, which can form an amorphous network structure in the lysosome called "proton sponge"^[4]. This unique

structure increases the lysosomal pH through protonation, prompting an ATP-driven influx of protons and chloride ions to restore the acidic environment within the lysosome. Subsequently, water influx counteracts the resulting osmotic imbalance, causing swelling and rupture of the lysosomal compartment, leading to the release of endosomes into the cytoplasm^[5]. The endocytosis pathway of P-MSN/miRNA nanoparticles was studied *in vitro*. Cy3-labeled P-MSN/miRNA nanoparticles (red) were co-cultured with adherent NRVM cells, and the positions of P-MSN/miRNA nanoparticles and LysoTracker-labeled lysosomes (green) within the cells were examined using confocal microscopy at 1 hour, 3 hours, and 6 hours. Initially, a significant amount of yellow fluorescence was observed in the cells at 1 hour, indicating the presence of aggregates within the lysosomes where green and red fluorescences overlapped. By 3 hours, the yellow fluorescence diminished, with an increase in separate green and red fluorescence signals, suggesting the nanoparticles were starting to disengage from the lysosomes. Subsequently, at 6 hours, the yellow fluorescence disappeared, leaving only distinct green and red fluorescence signals in the cells, indicating complete separation of the nanoparticles from the lysosomes. Quantification using ImageJ software revealed a substantial co-localization of nanoparticles and lysosomes at 1 hour, decreasing overlap at 3 hours, and minimal overlap at 6 hours, demonstrating successful escape of the nanoparticles from the lysosomes (Fig.6c).

To evaluate the integrity of microRNA after intracellular lysosomal escape, we examined miR199a-5p expression in primary cardiomyocytes for 12 hours following the introduction of P-MSN/miR199a-5p nanoparticles. RT-qPCR analysis revealed a progressive increase in miR199a-5p expression at 1, 3, 6, and 12 hours post-transfection with the nanoparticles (Fig. S4e). These results strongly indicate the preservation of miR199a-5p integrity, highlighting the delivery system's efficacy in preserving microRNA stability within the cellular environment.

FigS4e

- [1] BEYTH N, HOURI-HADDAD Y, BARANESS-HADAR L, et al. Surface antimicrobial activity and biocompatibility of incorporated polyethylenimine nanoparticles [J]. *Biomaterials*, 2008, 29(31): 4157-63.
- [2] XIANG S D, SCHOLZEN A, MINIGO G, et al. Pathogen recognition and development of particulate vaccines: does size matter? [J]. *Methods*, 2006, 40(1): 1-9.
- [3] DOBROVOLSKAIA M A, MCNEIL S E. Immunological properties of engineered nanomaterials [J]. *Nat Nanotechnol*, 2007, 2(8): 469-78.
- [4] MERDAN T, KUNATH K, FISCHER D, et al. Intracellular processing of poly(ethylene imine)/ribozyme complexes can be observed in living cells by using confocal laser scanning microscopy and inhibitor experiments [J]. *Pharm Res*, 2002, 19(2): 140-6.
- [5] AKINC A, THOMAS M, KLIBANOV A M, et al. Exploring polyethylenimine-mediated DNA transfection and the proton sponge hypothesis [J]. *J Gene Med*, 2005, 7(5): 657-63.

4. What is the half-life of miR199a-5p? For a single-dose but long-term therapeutic, either the delivered molecule or the regulated cascade has the potential of lasting effects. This reviewer also does not think the revealed mechanistic targets have lasting regulatory potential. No epigenetic or metabolic research has been performed.

Responses: Thanks for your advice. The literature reports an average microRNA half-life of 119 hours after correcting for cell division^[1]. To assess the persistence of our delivery system in cardiomyocytes, we transfected P-MSN/miR-cy3 and P-MSN/miR199a-5p nanoparticles into primary cardiomyocytes. Over a 4-week period, we detected both cy3 fluorescence and miR199a-5p levels intracellularly. The

findings revealed sustained cy3 fluorescence in cardiomyocytes for up to 4 weeks, and peak miR199a-5p levels observed after 1 week and continued levels for over 4 weeks (Fig.S4i, j). These results indicate that P-MSN/miR199a-5p nanoparticles can deliver and release miR199a-5p steadily in cardiomyocytes for an extended period. Additionally, in vivo experiments demonstrated a significant elevation of miR199a-5p levels in the infarct zone compared to the distal region after 4 weeks of tail vein injection of P-MSN/miR199a-5p nanoparticles in rats (Fig.S5a), aligning with the in vitro findings.

Based on the available results, the realization of long-term infarct repair effects of P-MSN/miR199a-5p nanoparticles in vivo may be attributed to the following. First, P-MSN/miR199a-5p nanoparticles significantly increased miR199a-5p within 1 week and slow-released miR199a-5p for at least 4 weeks, which steadily increased miR199a-5p expression within cardiomyocytes. Second, analysis of miRNA expression at different time points after infarction in the GEO database showed that miR199a-5p expression decreased significantly within 1 weeks after infarction, and the low expression of miR199a-5p within 1 week might be associated with sudden cardiac death, whereas miR199a-5p expression could increase slowly and with individual variability after 4 weeks. Therefore, P-MSN/miR199a-5p nanoparticles are adapted to the changing rule of miR199a-5p expression during the pathophysiological process of myocardial infarction, which decreases significantly in the early stage of miR199a-5p expression and increases slowly in the middle and late stages of myocardial infarction, so that the expression of miR199a-5p can be increased significantly in the first week of infarction in order to fulfill the function of infarction repair, and then increased slowly in the middle and late stages of infarction to avoid unpredictable side effects of the long-term expression for large quantities of miR199a-5p.

- [1] GANTIER M P, MCCOY C E, RUSINOVA I, et al. Analysis of microRNA turnover in mammalian cells following Dicer1 ablation [J]. *Nucleic Acids Res*, 2011, 39(13): 5692-703.

FigS4i

FigS4j

FigS5a

Minor concerns:

1. Fluorescence in situ hybridization should be performed on heart tissues for confirming the microRNA expression.

Responses: We thank the reviewers for this suggestion. In accordance with the suggestion, we examined the levels of miR199a-5p on heart tissues using the FISH technique, and the results showed that in the rat infarct region, miR199a-5p expression was increased in the P-MSN/miR199a-5p group compared with the negative control group (Fig.S5d).

FigS5b

2. Histological examinations should be performed to better show the details of infarcted tissues. IF staining of certain protein markers should also added.

Responses: Thanks for your advice. Additional Hematoxylin and Eosin (H & E) staining experiments were conducted on the infarcted areas of rats in each group four weeks post-infarction. The results revealed that the MI and P-MSN/NC+MI groups exhibited a predominant presence of scar tissue with minimal myocardial tissue visible under the microscope, along with residual inflammatory cell infiltration. Conversely, the MSN/miR199a-5p group displayed a reduction in scar tissue accompanied by increased myocardial tissue. Notably, the P-MSN/miR199a-5p group demonstrated the least amount of scar tissue and the highest amount of myocardial tissue, indicative of enhanced repair effectiveness (Fig.S5e). In addition, we examined the beating myocardium marker protein(actinin), and immunofluorescence assay demonstrated a significant increase in actinin expression in the P-MSN/miR199a-5p group compared with the P-MSN/NC+MI group (Fig.7d).

FigS5e

3. Would miR199a-5p be released by cells in forms of extracellular vesicles? Please discuss any potential side-effects or factors affecting the robustness of the

methodology.

Responses: The literature indicates that miR199a-5p can be released by cells in the form of extracellular vesicles. Studies suggest that bone marrow mesenchymal stem cells protect against renal ischemia-reperfusion injury by releasing exosomes containing miR-199a-5p^[1]. Additionally, hepatic stellate cells secrete exosomes loaded with miR-199a-5p to target connective tissue growth factor (CCN2) and inhibit fibrotic signaling^[2]. Furthermore, endothelial cell-derived exosomes are reported to promote and sustain a repair-associated phenotype of Schwann cells through miR199a-5p, thus facilitating nerve regeneration^[3].

When considering potential side effects or factors affecting the robustness of the methodology, we take into account the following: Firstly, despite the clear support from multiple delivery systems in the literature indicating that PEP peptides targeting injured myocardium have low off-target properties, in clinical application, increasing the dosage of the delivery system to achieve therapeutic effects may still result in a high concentration of miRNA in the circulation, leading to unknown adverse reactions. Non-invasive in situ delivery may be a preferable choice. Secondly, despite the recognized benefits of microRNA and the overall biosafety and biocompatibility of silica-based nanoparticles, it is important to note that these nanoparticles have a slow degradation rate and relatively long degradation time. To address this limitation, exploring alternative delivery carriers that are easy to design and prepare can be beneficial, ensuring smoother translational applications in the future.

- [1] WANG C, ZHU G, HE W, et al. BMSCs protect against renal ischemia-reperfusion injury by secreting exosomes loaded with miR-199a-5p that target BIP to inhibit endoplasmic reticulum stress at the very early reperfusion stages [J]. *Faseb j*, 2019, 33(4): 5440-56.
- [2] CHEN L, CHEN R, VELAZQUEZ V M, et al. Fibrogenic Signaling Is Suppressed in Hepatic Stellate Cells through Targeting of Connective Tissue Growth Factor (CCN2) by Cellular or Exosomal MicroRNA-199a-5p [J]. *Am J Pathol*, 2016, 186(11): 2921-33.
- [3] HUANG J, ZHANG G, LI S, et al. Endothelial cell-derived exosomes boost and maintain repair-related phenotypes of Schwann cells via miR199-5p to promote nerve regeneration [J]. *J Nanobiotechnology*, 2023, 21(1): 10.

REVIEWERS' COMMENTS

Reviewer #1 (Remarks to the Author):

The authors have responded comprehensively to my concerns with edits to the text, figures, and new experiments which now more robustly support particular conclusions.

The primary conclusions, which are novel and significant, appear appropriately supported by the data.

Reviewer #2 (Remarks to the Author):

The authors have addressed the majority of points raised in my review.

1. The authors have now provided more information in relation to the reproducibility of the MI procedure via LAD. This information should be added to the methods.
2. It is noted that echocardiography data has been provided in Table 1 and Table 2. I found raw echo data in Supp Table 1 and 2 for individual animals. I did not find Table 1 and Table 2? In addition to the raw data for individual animals, the mean data should be presented for each group in a table as requested.
3. The authors note that Western blot images are provided in the raw data file. I was unable to find these images.

Reviewer #3 (Remarks to the Author):

The authors have made required revisions and explanations. The concerns are addressed.

REVIEWERS' COMMENTS

Thank you for your comments on our manuscript. We appreciate the valuable feedback you provided concerning our work. In response to these comments, we have made direct modifications to the text and the table. We believe that the revised manuscript adequately addresses and incorporates all the issues raised by the reviewers during the resubmission of the manuscript. The following are the specific modifications we have made in response to the feedback from all reviewers.

Reviewer #1 (Remarks to the Author):

The authors have responded comprehensively to my concerns with edits to the text, figures, and new experiments which now more robustly support particular conclusions.

The primary conclusions, which are novel and significant, appear appropriately supported by the data.

Responses: **We thank the reviewers for their commendation on the breadth and depth of our research, and appreciate their time spent reviewing and re-reviewing our manuscript.**

Reviewer #2 (Remarks to the Author):

The authors have addressed the majority of points raised in my review.

1. The authors have now provided more information in relation to the reproducibility of the MI procedure via LAD. This information should be added to the methods.

Responses: **Thanks for your advice. In this revised version, we have included comprehensive details on the reproducibility of myocardial infarction procedures involving the LAD in the Methods section.**

2. It is noted that echocardiography data has been provided in Table 1 and Table 2. I found raw echo data in Supp Table 1 and 2 for individual animals. I did not find Table 1 and Table 2? In addition to the raw data for individual animals, the mean data

should be presented for each group in a table as requested.

Responses: We regret any confusion that may have arisen. Supplementary Table 1 and Table 2 are the same as Table 1 and Table 2. We will rename Table 1 and Table 2 and add mean data to the tables.

3. The authors note that Western blot images are provided in the raw data file. I was unable to find these images.

Responses: We deeply apologize for the mistake and the inconvenience caused. We will upload the source data files in Excel format, including all uncropped scans of all blots and gels.

Reviewer #3 (Remarks to the Author):

The authors have made required revisions and explanations. The concerns are addressed.

Responses: We thank the reviewer for their approval of our revision after taking the time to re-read it